# Remapping the foundations of morality: Well-fitting structural model of the Moral Foundations Questionnaire

**Michael Zakharin**⬥*, **Timothy C. Bates**

Department of Psychology, University of Edinburgh, Edinburgh, United Kingdom

* s1775682@sms.ed.ac.uk

## Abstract

Moral foundations theory posits five moral foundations, however 5-factor models provide poor fit to the data. Here, in five studies, each with large samples (total N = 11,496), we construct and replicate a well-fitting model of the Moral Foundations Questionnaire (MFQ). In study 1 (N = 2,271) we tested previously theorised models, confirming none provide adequate fit. We then developed a well-fitting model of the MFQ. In this model, the fairness/reciprocity and harm/care foundations were preserved intact. The binding foundations, however, divided into five, rather than the original three foundations. Purity/sanctity split into independent foundations of purity and sanctity. Similarly, Ingroup/loyalty divided into independent factors of loyalty to clan and loyalty to country. Authority/respect was re-focussed on hierarchy, losing one item to the new sanctity foundation and another into loyalty to country. In addition to these 7 foundations, higher-level factors of binding and individualizing were supported, along with a general/acquiescence factor. Finally, a "moral tilt" factor corresponding to coordinated left-leaning vs. right-leaning moral patterns was supported. We validated the model in four additional studies, testing replication of the 7-foundation model in data including from US, Australia, and China (total N = 9,225). The model replicated with good fit found in all four samples. These findings demonstrate the first well-fitting replicable model of the MFQ. They also highlight the importance of modelling measurement structure, and reveal important additional foundations, and structure (binding, individualizing, tilt) above the foundations.

## Introduction

Moral psychology seeks to understand how people form moral judgments, why individuals differ in these judgments and what the structure of these judgements is [1]. Moral Foundations Theory [MFT: 2] has emerged as the leading theory addressing these questions, suggesting that moral judgment arises from five universal foundations, each evolved to facilitate cooperative relations [2]. The Moral Foundations Questionnaire [MFQ: 3] was developed based on this theory and has been used in many hundreds of studies. To date, however, no well-fitting model of the MFQ has been produced [4, 5]. Here, we test the fit of existing models, confirming that these fit poorly. We then used a multi-trait multi-method approach to develop a well-

**Data Availability Statement:** The data and materials used in this paper as well as R code used to generate the results are openly available at the OSF site for this paper at https://osf.io/bt9nh/, DOI 10.17605/OSF.IO/BT9NH.

**Funding:** The authors received no specific funding for this work.

**Competing interests:** The authors have declared that no competing interests exist.

fitting model of the MFQ, and demonstrate that this new model replicates in multiple large independent samples. Below, we briefly introduce the foundations, the MFQ measurement of these foundations, and outline previous tests of the model and proposed alternative models before presenting study 1, which develops a well-fitting model of the MFQ items.

Applying an evolutionary approach to previous theories of morality and human values [6–9], Haidt and colleagues argued that morality consists of at least five culturally universal moral domains identified as follows: First, *harm/care* concerns avoidance of suffering for others and is experienced as compassion. Second, *fairness/reciprocity* concerns avoidance of unfairness or unequal treatment of self and others and failure to reciprocate. When triggered, it is experienced as personal anger if one is oneself the target of unfair action or being taken advantage of, or empathic anger if the target is another person. Third, *ingroup/loyalty* concerns maintenance of loyalty to one's group by oneself and by other members of one's group. When triggered, this foundation is experienced positively as feelings of unity with the members of one's group, as feelings of guilt if one is tempted to betray one's group and as feelings of treachery if others are disloyal. Fourth, a*uthority/respect* is concerned with recognizing, respecting and preserving societal hierarchy. It is triggered by the presence of hierarchy cues and by behavior lacking respect for hierarchy. When activated, it is reflected in feelings of deference and respect for the authority of these hierarchies. Patriotism or loyalty to country has been contrasted by Haidt [10] with globalism, creating a dimension of caring for the people in one's own country more than those of other countries, or treating all people as identical in terms of their moral call upon us.

Finally, *purity/sanctity* is a foundation proposed to have evolved both to avoid toxic and parasitic contamination, but also to promote beliefs and ritual: what Durkheim identified as "*sacred things. . . things set apart and forbidden*" [11/1912, p. 44]. Haidt and Graham (2) describe purity as "*a guardian of the body in all cultures, responding to elicitors that are biologically or culturally linked to disease transmission*" (p. 106). Separately, ritualised beliefs contrast virtues where "*the soul is in charge of the body*" against unnatural vices such as lust and gluttony seen as "*debased, impure, and less than human*" (p. 106). In each culture, morally impure practices violating purity/sanctity are experienced as feelings bearing some similarity to disgust and revulsion [2]. Finally, we should note that liberty was also identified as a possible moral foundation [12], but is typically omitted in questionnaire studies of moral foundations theory. Following Graham, Haidt [13], it is common to group the five moral foundations into two super-factors based on whether the locus of moral value is the individual or the group. In this scheme, the foundations of harm/care and fairness/reciprocity are grouped as "individualising" foundations and the foundations of ingroup/loyalty, authority/respect and purity/sanctity are categorized as "binding" foundations because their locus of moral value is the group. The relative strength of the individualising versus the binding foundations in a particular individual are argued to underpin individual differences between liberal and conservative values [13–15].

A strength of moral foundations theory is that the authors created an open measurement instrument, allowing others to test the structure and predictions of the moral foundations model. The MFQ-30 [3] consists of 30 items plus 2 foil items used to filter out inattentive participants. It also includes two distinct item formats: a 15 "relevance" items section and a 15 "judgment" items section containing three items for each of the five foundations. The inclusion of two measurement methods within the questionnaire is an under-exploited strength, allowing improved modelling accuracy [16]. Items in the "relevance" block measure the moral relevance of various aspects of behavior, posing an example, e.g. "*Whether or not someone was cruel*" and asking participants to score how relevant to them this behavior would be in reaching a moral decision from "not at all relevant" to "extremely relevant". By contrast, judgment items assess the extent to which participants agree with a specific moral judgment, e.g. "*I am*

*proud of my country's history*" scored strongly agree to strongly disagree). All MFQ-30 items are measured on a 6-point Likert scale.

Since its development, the correlation of the MFQ with external measures has been widely studied, especially in the domain of politics. For example, moral foundations scores predict voting outcomes over and above traditional demographic predictors [17] and MFQ scores correlate with location on the right wing-left wing ideological divide [13, 18], as well as people's stances on other 'culture war' attitudes [19]. Fairness, authority, ingroup and purity account for significant variance in the self-identification with distinct religious orientations postulated by religious orientations theory [20]. In sacrificial dilemma studies, harm, purity and ingroup foundations are associated with endorsement of harmful action [21]. There has also been support for brain volumes being associated with the MFQ responses [22].

Alongside this supportive research, both moral foundations theory and the MFQ measure have been criticised both on theoretical and empirical grounds (e.g. by papers using it as a trait of interest). Regarding the prediction of political orientation, in a meta-analysis of the relationship between moral foundations and political orientation, Kivikanagas et al. [18] found that correlations between the five foundations and political orientation were close to zero in Black samples. At the level of taxonomy, Suhler and Churchland [23] have questioned the basis for the selection of the five foundations as foundational. In a similar vein, Curry et al. [24] argued that the theory includes content they consider non-moral (purity/sanctity and harm/and care), collapses domains which in the view of Curry et al. [24] should be kept distinct, and misses entirely other moral domains (e.g. heroism). Smith et al. and Hatemi et al. [25, 26], using multiple samples, reported that the MFQ did not reliably factor into 5 dimensions, but rather into 2, with structure differing between the US and Australia. They also found that moral foundations are not stable across time and that MFQ scores reflect rather than cause political attitudes. Smith et al. [25] also reported that the MFQ foundations show no evidence of heritability. Regarding the foundational nature of the moral foundations, Strupp-Levitsky et al. [27] suggest that rather than being causal, the foundations build on other, more basic, variables such as empathy, need for closure and need for cognition. In a study manipulating partisan and group identity cues by embedding these in modified MFQ items, Ciuk [28] reported that item endorsement was affected by partisan alignment, supporting the conclusion that causality runs from political ideology to moral foundations. At the psychometric level, Iurino and Saucier [5] tested measurement invariance of the 5-factor structure of the MFQ in 27 countries and concluded that there was little support for a five-factor solution for the questionnaire. Similarly, in a US sample, Davis et al. [29] tested measurement invariance of the MFQ in Black and White samples, concluding that the assumption of scalar invariance could not be supported. Jointly, these pose considerable challenges for a measure that aims to be culturally universal, perhaps especially problems in finding a well-fitting model as a basis for prediction.

We next turn to the measurement implementation of moral foundations theory in the MFQ, reviewing previous work testing predicted theoretical structures. Specifically, we address the question of whether the MFQ shows the proposed structure of (at least) five distinct moral foundations aligned with the predicted item content. We then build and test a series of models, testing existing five foundation models, and, after showing that this does not fit, exploring alternative formulations improving measurement modelling of the MFQ, before replicating this model in a series of independent datasets.

## Psychometric modelling of the Moral Foundations Questionnaire (MFQ)

In their seminal paper, Graham et al. [3] collected online responses from a large international sample, mostly from Western countries (total N = 34,476). They tested one to six-factor

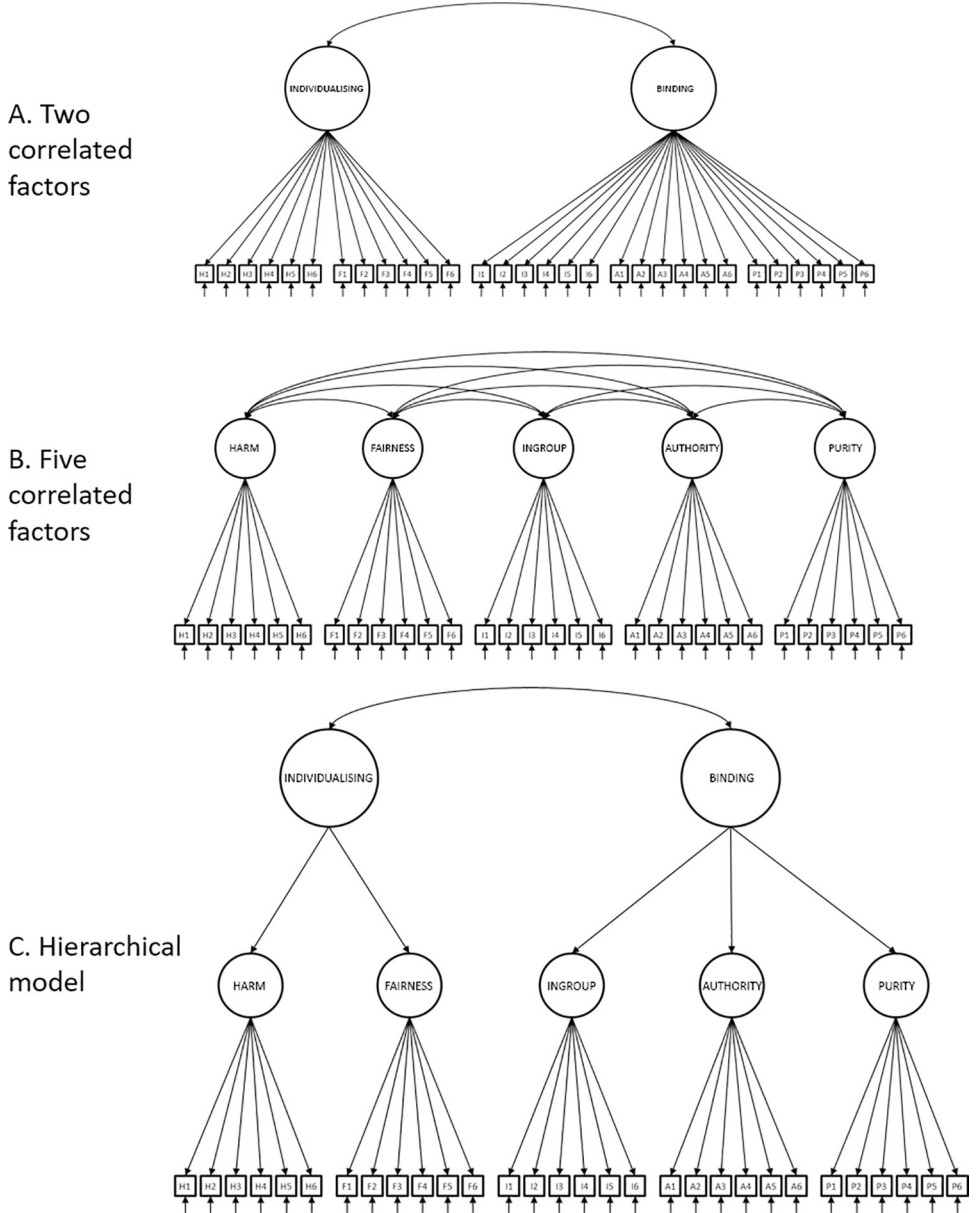

**Fig 1.** The three models previously applied to the MFQ: A: 2-factor model, B: 5-factor model, C: Hierarchical model.

models as well as a hierarchical five-factor model with two super-factors. Fig 1 shows the structure of two-factor, five-factor, and hierarchical models. The best fitting model consisted of five correlated factors. This model fit better than competing models and showed adequate fit according to Root Mean Square Error of Approximation (RMSEA) = .046. However, by other important conventional indices, the model fit poorly. For instance, the Comparative Fit Index (CFI) for this model was only .824, well below the generally accepted value of 0.95 [30]. No reports, to our knowledge, have resulted in satisfactory fit (see Table 1 for a sample of fits for different MFQ models in different samples and cultures).

Since the initial work by Graham et al. [3], several reports have been published aimed at replicating or improving the five-factor structure of the MFQ. Davies et al. [4] applied the

**Table 1. MFQ fit metrics obtained in the previous studies.**

| Study | N | Metrics reported | | MFQ version | Best fitting model | Sample description |
|---|---|---|---|---|---|---|
| | | RMSEA | CFI | | | |
| Curry et al. (2019) | 1,042 | 0.050 | 0.910 | MFQ-30 | 7 factors | UK online sample |
| Graham et al. (2011) | 413 | 0.043 | 0.876 | MFQ-30 | 5 factors | Eastern Europe online sample |
| Iurino & Saucier (2018) | 8,055 | 0.084 | 0.853 | MFQ-20 | 5 factors (modified) | Survey of World Views international sample |
| Yalçındağ et al. (2019) | 1432 | 0.104 | 0.850 | MFQ-30 | 5 factors | Three Turkish samples |
| Graham et al. (2011) | 411 | 0.039 | 0.841 | MFQ-30 | 5 factors | Latin America online sample |
| Graham et al. (2011) | 299 | 0.042 | 0.838 | MFQ-30 | 5 factors | South Asia online sample |
| Davies et al. (2014) | 3994 | 0.063 | 0.829 | MFQ-30 | 5 factors | New Zealand national probability sample |
| Graham et al. (2011) | 26,014 | 0.048 | 0.824 | MFQ-30 | 5 factors | US online sample |
| Graham et al. (2011) | 1,670 | 0.046 | 0.811 | MFQ-30 | 5 factors | Western Europe online sample |
| Kivikangas et al. (2017) | 874 | 0.078 | 0.749 | MFQ-30 | 5 factors | Finnish nationally representative sample |
| Ji & Janicke (2018) | 234 | 0.077 | 0.744 | MFQ-30 | Hierarchical | Chinese students |
| Kim, Kang & Yun (2012) | 478 | 0.068 | 0.681 | MFQ-30 | 5 factors | South Korean students |
| Nilsson & Erlandsson (2015) | 540 | 0.072 | 0.679 | MFQ-30 | 5 factors | Swedish students |
| Ji & Janicke (2018) | 204 | 0.078 | 0.658 | MFQ-30 | Hierarchical | US students |
| Harper & Rhodes (2021) | 322 | 0.080 | 0.77 | MFQ-30 | 5 factors | UK online sample |
| Hadarics & Kende (2017) | 403 | 0.091 | 0.681 | MFQ-30 | 5 factors | Hungarian student sample |
| Doğruyol et al. (2019) | 4,971 | 0.050 | 0.940 | REL-15 | 5 factors | Many Labs 2 Project, Western countries |
| Doğruyol et al. (2019) | 1,997 | 0.060 | 0.940 | REL-15 | 5 factors | Many Labs 2 Project, Non-Western countries |
| Du (2019) | 761 | 0.080 | 0.930 | REL-15 | 5 factors | Two Chinese samples |

MFQ-20 is the short 20-item version of MFQ. Rel-15 is the 15-item Relevance subscale from MFQ-30. Iurino & Saucier (2018) used alternative model derived from exploratory factor analysis of original MFQ items

models tested by Graham et al. [3] to new data collected in New Zealand (N = 3,994). They concluded that the 5-factor model fit better than models with more or fewer factors but that fit metrics, specifically the CFI, were unsatisfactory. Nilsson & Erlandsson [31] modified the hierarchical model, separating purity from other binding foundations and testing this in a sample of Swedish students (N = 540). However, they found that a five correlated-factors models showed the best fit in their sample despite, again, no model showing adequate fit. More recently, Harper & Rhodes [32] tested the factor structure of the MFQ in two British samples (total N = 750), confirming that the proposed five-factor structure was not psychometrically sound according to accepted metrics. They also tested an extended MFQ, including the nine items of the sixth "Liberty" foundation proposed by Haidt and colleagues [12]. Adding the Liberty scale, however, did not lead to a well-fitting six-factor model, and instead was better explained by a three-factor model comprising "traditionalism", "compassion" and "liberty".

The structure of the MFQ has also been investigated in non-Western samples. Yalçındağ et al. [33] used three Turkish samples to test replication of the models reported by Graham et al. [3]. In each sample, the 5-factor model was best-fitting, but no model had adequate fit. Hadarics and Kende [34] tested the 5-factor structure in a sample of Hungarian students, finding this model fit poorly. Iurino and Saucier [5] used the 20-item MFQ administered in the Survey of World Views [35] and covering respondents from 27 countries (N = 8055). An exploratory factor analysis supported a 5-factor model, though the item-factor loadings differed substantively from those proposed by Graham et al. [3] and formal fit of the model was below the threshold for acceptability.

Some researchers have tested models using just the relevance or the judgment items alone. Both formats yield similar structures. For example, Doğruyol et al. [36] compared the fit of a

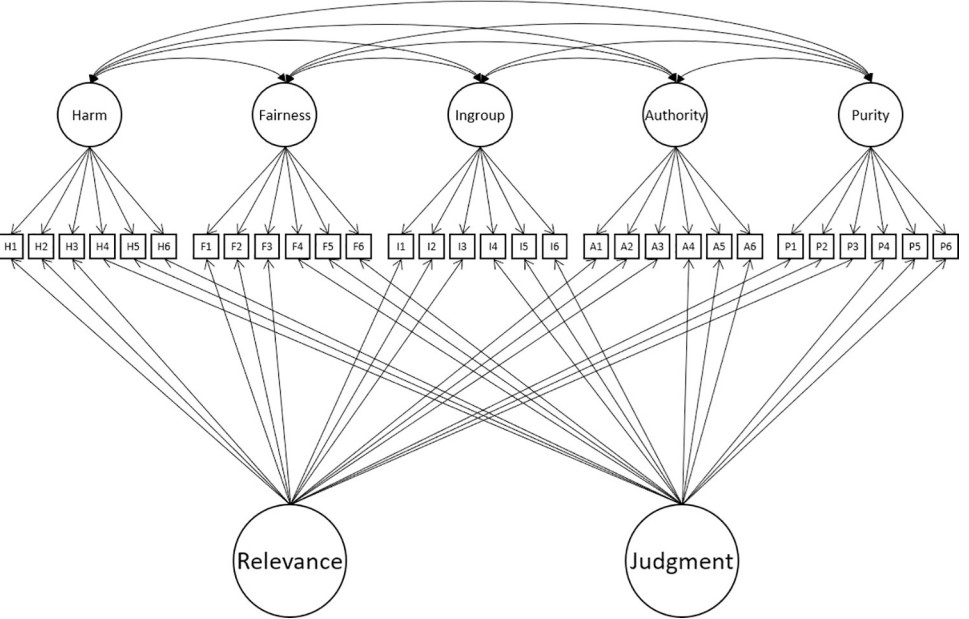

**Fig 2. Incorporating multitrait-multimethod into a 5-foundation model of the MFQ.**

5-factor and hierarchical models in relevance items from the many labs project [37]. They found that in both Western and non-Western countries the 5-factor model fit better than did a hierarchical model. Du [38] fit models with 1 to 5 factors in Chinese sample (N = 761), finding that the 5-factor model offered the best fit compared to models with fewer factors. Interestingly, although the MFQ authors included these distinct measurement methods (the relevance and judgment items), to our knowledge only one analysis has used these to account for measurement variance in a model. Curry et al. [24] added relevance and judgment method factors to an item-level 5-correlated factor model (see Fig 2) reporting that this improved fit. Still, though, this model did not meet the standard levels of acceptable fit as measured by CFI, leading Curry et al. [24] to reject the MFQ as failing to show a psychometrically valid measurement model.

## Summary and directions for improved modeling

Taken together, earlier attempts to model the MFQ-30 suggest that it is best modelled as including at least five factors, one for each of the proposed foundations. The imperfect fit of this model, however, indicates that, while a five-factor model may reasonably be considered as a good starting point to develop a better fitting structure of MFQ-30, significant elements of covariance structure are not captured by this model.

At least three plausible explanations could account for the poor fit of models tested to date. One is that correlated-factor models cannot readily represent clustering within the foundations, thus failing to fully reflect binding and individualising effects which form part of moral foundations theory. Supporting this, Graham et al. [3] modelled binding and individualising as hierarchical superfactors which improved the model, but not to the level considered a good fit. An alternative approach, not attempted to date, would involve implementing the binding and individualizing factors in a bi-factor structure. This also raises the possibility of implementing the model at the level of the items (see Fig 3). To our knowledge, there have been no reported attempts to model the group factors in bi-factor structures in item-level models.

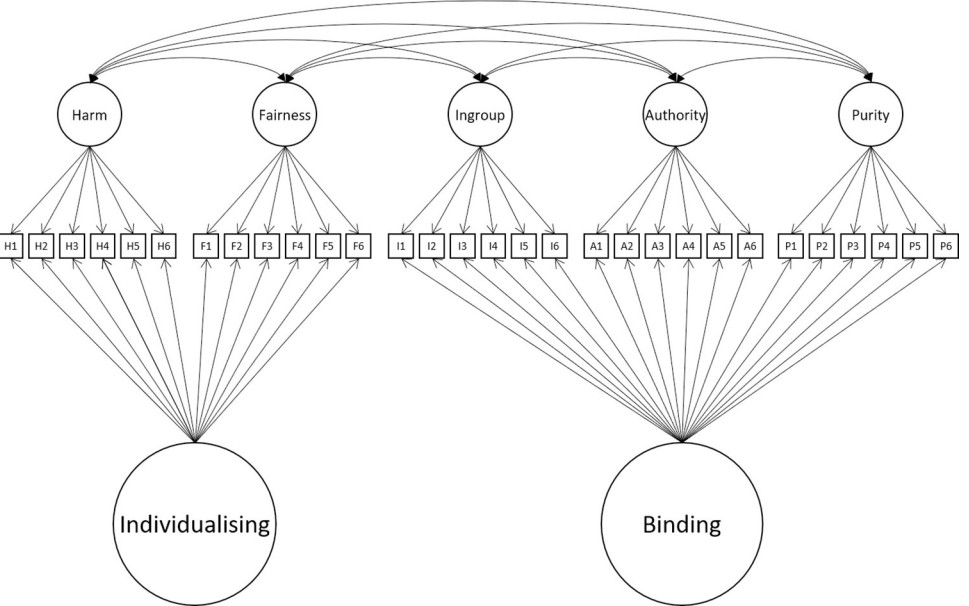

**Fig 3. 5-factor item-level model of including individualising and binding domains.**

A second explanation for poor fit may lie in the foundations themselves. One or more foundations may have sub-components which need to be modelled as distinct entities. Other foundations may be better modelled if collapsed together. Hence, splitting and collapsing of factors may be required. Third, and finally, general influences on responding, whether from a general morality factor, or response biases such as acquiescence [39] or social desirability [40] will cause item-covariance lowering the fit of models not accounting for these effects. It will be valuable, therefore, to explore the impact of these effects when building a psychometrically well-fitting model of the MFQ. To our knowledge, no previous attempts to model the MFQ have investigated these types of hypothesis even though this has been explored in personality models, where, for instance, social desirability has proven to be a useful measurement component of well-fitting models [41].

**Summary.** In the present paper, we used structural equation modelling to test existing models of the MFQ-30, and to build and validate a new, well-fitting structural model of this measure. In **Study 1,** we tested 4 MFQ models suggested in previous research. Specifically, we compared the fit of the two-factor, five-factor, hierarchical and Multitrait-Multimethod models in a novel dataset. We showed that a Multitrait-Multimethod model fits better than the three other designs but that no model fit well. Next, we developed a new model of the MFQ. Our approach was to develop models increasing in complexity from the simplest predicted model to more complex structures, as required to achieve good fit. Using Multitrait-Multimethod model as the base model, we found that including binding and individualizing factors were required, and that modeling these at the item level results in better fit compared to including them as associations among foundations. Two additional foundations, loyalty to country and purity (separated from sanctity) were required. Finally, we showed that adding a general morality and left-right tilt factors improved fit and that together, these innovations yielded a well-fitting model. In **Studies 2 through 5** we replicated the model developed in the Study 1 in four independent open-access samples showing that the model proposed and replicated internally in study 1 fits better than any other competing model in each of these datasets

and meets accepted criteria for good fit. The five samples used were, for Study 1, 2,271 students from the University of Edinburgh and members of the local community; for Study 2, 7,130 participants from Graham et al., study 3 [13]; for study 3, 1,052 participants from Smith et al. [25]; for Study 4, 553 participants from O'Grady et al. [42]; and for Study 5, 452 participants from Wang et al. [43]. Together, the studies amount to a total N of 11,496 and 4 independent replications.

## Study 1

### Method

**Participants.** A total of 2,271 people from the UK participated in the study. We used two attention check questions to remove inattentive participants: (a) "It is better to do good than to do bad" and (b) "It is relevant to moral judgment whether or not someone was good at math". Participants who responded with *slightly*, *moderately* or *strongly disagree* to the first question and *somewhat*, *very* or *extremely relevant* to the second question were excluded. The final sample consisted of 2,039 adults (1404 females, 631 males; age M = 25.3, SD = 13.04). Participants were recruited from a volunteer pool consisting of students and members of the community. After providing basic demographic data, participants completed the MFQ-30 online.

**Measures.** The extent to which participants endorsed moral foundations was measured by the 30-item Moral Foundations Questionnaire (MFQ-30) [3].

### Analytic approach

All analyses were conducted at the item level. We began by implementing and testing fit of 4 theoretical structural models reported in earlier research namely the 2-factor, 5-factor, hierarchical model and Multitrait-Multimethod model (see Figs 1 and 2). Model fit was assessed using the Comparative Fit Index (CFI), Tucker-Lewis Index (TLI) and the root mean square error of approximation (RMSEA). The comparative fit of the models was assessed by the Akaike Information Criterion (AIC) [44] which penalises un-parsimonious models. Following Hu & Bentler [30] and Yu [45] we adopted criteria of TLI > = .95 and RMSEA < = .06.

After examining the fit of previously theorised models of MFQ we turned to building a well-fitting model. The full sample was randomly split into training (N = 1,020) and holdout (N = 1,019) datasets. New models were built in the training dataset and the models achieving adequate fit were then tested in the holdout dataset. We used the Multitrait-Multimethod model as the base model for these new analyses. Modification attempts began with modelling binding and individualising factors within a bi-factor framework (i.e., these factors loaded directly on the items, rather than on the MFQ domain factors (see Fig 3). This choice was made a-priori based on research suggesting that that structuring these hierarchically does not improve model fit [3, 4, 24, 31, 33]. Paths from both binding and individualizing factors to their corresponding items were constrained to be positive to ensure that these factors could capture only variance common to all items within their respective clusters. Second, we investigated whether the model can be improved by changing the numbers of foundations. Finally, we investigated whether adding one or more general factors would improve fit in the model. There are several theoretical reasons why general factor can be expected to be found in a personality measurement scales such as MFQ. First, general factor can represent a genuine factor of personality. Second, general factor can arise as a result of acquiescence bias, tendency to agree with all questionnaire items [39]. Finally, self-enhancement or social desirability effect, the tendency to overestimate one's positive personality traits can also manifest itself as a general factor.

**Table 2. Study 1 comparative model fits, reported in order of most complex to least complex.**

| Model | EP | Δ -2LL | Δ df | p | AIC | Compare with Model |
|---|---|---|---|---|---|---|
| **1. Multitrait-multimethod** | **130** | | | | **65049.98** | |
| 2. Hierarchical model | 101 | 1913.56 | 29 | < .001 | 66905.54 | Multitrait-multimethod |
| 3. 5-factor model | 100 | 1848.69 | 30 | < .001 | 66838.67 | Multitrait-multimethod |
| 4. 2-factor model | 91 | 2244.61 | 39 | < .001 | 67216.59 | Multitrait-multimethod |

AIC = Akaike information criteria. Low AIC values indicate better fit. Best fitting model is printed in bold.

## Results

All statistical analyses were completed in R [46] and umx [47].

We first tested the hypothesis that binding and individualizing factors are sufficient to explain the variance in the MFQ (2-factor model). This resulted in unsatisfactory fit, $\chi^2$ (404) = 5148.24, p < 0.001; CFI = 0.699; TLI = 0.676; RMSEA = 0.076. Next, we tested whether a five-factor model would offer better fit to the data. This also resulted in unsatisfactory fit ($\chi^2$ (395) = 4752.32, p < 0.001; CFI = 0.724; TLI = 0.696; RMSEA = 0.074). Then, we tested whether adding two superfactors to the five-factor model (the hierarchical model) would improve the fit. This was not the case ($\chi^2$ (394) = 4817.19, p < 0.001; CFI = 0.719; TLI = 0.69; RMSEA = 0.074). Finally, we tested Multitrait-Multimethod model. The fit of this model was better than the fit of three other models, however the fit statistics (with the exception of RMSEA) were still below the acceptable level: $\chi^2$ (365) = 2903.63, p < 0.001; CFI = 0.839; TLI = 0.808; RMSEA = 0.058. Although none of the models achieved acceptable fit, the Multitrait-Multimethod model offered the best fit to the data (see Table 2 for the model comparisons).

After confirming that existing proposed models did not fit adequately, and that a Multitrait-Multimethod model fit better than the alternatives, we proceeded to attempt to develop a better-fitting model based on this foundation. The fit statistics accompanying each step we took to improve the Multitrait-Multimethod model fit in the training dataset are shown in Table 3.

The first change made was to include factors representing binding and individualising foundations at the item level (Model 2). This led to a significant improvement in fit of the Multitrait-Multimethod model ($\chi^2$(334) = 1111.43, p < 0.001; CFI = 0.901; TLI = 0.871; RMSEA = 0.048). Next, we examined the residuals (unmodeled covariance among items) in this model. This indicated that two groups of items had covariance not accounted for by their membership of a foundation nor by broader binding or individualising factors, suggesting a need for two additional factors. To model these groupings, two additional factors were added. The first split the items in the sanctity/purity foundation, dividing these among a factor

**Table 3. Model fit comparisons for the training dataset in Study 1.**

| Model | EP | CFI | TLI | RMSEA | AIC | Compare with Model |
|---|---|---|---|---|---|---|
| M1. Multitrait-Multimethod model | 130 | .843 | .813 | .057 | 32757.7 | Model 5 |
| M2. M1 + Binding & individualising | 161 | .901 | .871 | .048 | 32337.7 | Model 5 |
| M3. M2 + Two new foundations | 172 | .918 | .890 | .044 | 32212.9 | Model 5 |
| M4. M3 + One general factor | 202 | .950 | .926 | .036 | 31989.9 | Model 5 |
| **M5. M4 + Second general factors** | **232** | **.963** | **.939** | **.033** | **31918.2** | |

AIC = Akaike information criteria; Best fitting model is model 5, printed in bold.

loading on two sanctity/purity items ("*Whether or not someone acted in a way that God would approve of*" and "*Chastity is an important and valuable virtue*"), and one authority item "*Men and women each have different roles to play in society*". We assigned the name "sanctity" to this factor. The four remaining items from the original sanctity/purity foundation involve attitudes towards avoiding disgusting or unnatural things so we named this factor "purity". The second factor split off two loyalty items ("Whether or not someone's action showed love for his or her country" and "I am proud of my country's history") and one authority item ("*If I were a soldier and disagreed with my commanding officer's orders, I would obey anyway because that is my duty*"). We termed this factor "loyalty to country". Adding these two additional foundations (Model 3) significantly improved fit relative to Model 2 ($\chi^2$(323) = 964.64, p < 0.001; CFI = 0.918; TLI = 0.89; RMSEA = 0.044). The fit of this model, while improved, still, however, fell below modern standards.

Next, we explored the effects of adding a general factor to the model with paths to items allowed to load positively or negatively on this factor (see Model 4 in Table 3). Adding this unconstrained general factor improved fit compared to Model 3 ($\chi^2$(293) = 681.65, p < 0.001; CFI = 0.95; TLI = 0.926; RMSEA = 0.036). The general factor loaded positively on fairness and harm items and negatively on items related to authority and on purity. Example fairness loadings included "*I think it's morally wrong that rich children inherit a lot of money while poor children inherit nothing*" (ß = .35), "*Whether or not some people were treated differently than others*" (ß = .29;). The harm item "*Whether or not someone cared for someone weak or vulnerable*" loaded ß = .27. Negative loadings on authority items included "*Respect for authority is something all children need to learn*" (ß = -.54), "*Men and women each have different roles to play in society*" (ß = -.53) and "*I am proud of my country's history*" (ß = -.53). Loadings on *purity* items were smaller but all negative. From these loadings, we concluded that this factor measured a domain running from liberal egalitarianism to conservatism. This would be consistent with the characterisation within moral foundations theory of the difference in liberal and conservative orientation. Alternatively, this factor may reflect social dominance [48] or social desirability bias. However, we could not test this with the present data.

Next, we investigated whether modifying Model 4 by adding a second factor, in this case constrained to load positively on all items, would improve fit. Given all items are scored in the same direction, this factor modelled overall greater or lower levels of moral concern/orientation, possibly representing acquiescence bias. We added this additional general/acquiescence factor to Model 4, with all paths from this factor to the items constrained to be positive (Model 5). This change improved fit compared to the Model 4, and yielded the first well-fitting model by modern standards ($\chi^2$(263) = 552, p < 0.001; CFI = 0.963; TLI = 0.939; RMSEA = 0.033). Table 3 shows the comparison statistics for models 1 through 5.

Having generated a well-fitting model in the training dataset, we next tested replication of this model in the holdout dataset (N = 1,019) to test if the model is reliable, and not over-fitted. For additional confirmation we also tested each of the intermediate models generated in study 1 verifying that the improvements observed across each change also replicated in this independent sample, as well replicating as the final model.

The results of this replication in the hold-out sample closely replicated those from the test dataset. As can be seen in Table 4, each modification which improved model fit in the training set also improved model fit in the holdout dataset. Importantly, the final model (seven foundations model with two general factors that also accounts for two measurement methods as well as for binding and individualising domains)—fit well in the independent hold-out data. This strongly confirms the new model and raises confidence that the changes made in seeking a well-fitting model were not simply over-fitting noise in the initial dataset, but identifying important factors in the structure of moral foundations.

**Table 4. Model fit comparisons for the holdout dataset replication in Study 1.**

| Model | EP | CFI | TLI | RMSEA | AIC |
|---|---|---|---|---|---|
| M1. Multitrait-Multimethod model | 130 | .828 | .795 | .061 | 32416.76 |
| M2. M1 + Binding and individualising | 161 | .883 | .847 | .053 | 32004.8 |
| M3. M2 + Two new foundations | 172 | .901 | .866 | .049 | 31873.44 |
| M4. M3 + One general factor | 202 | .936 | .904 | .042 | 31621.77 |
| **M5. M3 + Second general factors** | **232** | **.956** | **.927** | **.036** | **31489.21** |

AIC = Akaike information criteria; Best fitting model is model 5, printed in bold.

## The final model

Having shown that the model developed in the original dataset replicated in the hold-out sample, we reproduced this model in the combined discovery and holdout datasets for maximum precision of estimation of the effect sizes in the model. This model fit well ($\chi^2$(263) = 829.19, p < 0.001; CFI = 0.964; TLI = 0.941; RMSEA = 0.032.). We present this model graphically to make clear the findings of study 1. For clarity the model is presented in two parts. Fig 4 shows the 7 foundations and two general factors identified in our final model. Fig 5 shows the measurement part of the model which includes group factors. Full details of the model are tabulated in the OSF site for this paper.

## Discussion

Study 1 yielded three important findings. First, replicating previous studies (e.g. [3, 4, 33]), simpler models did not fit well: neither hierarchical model, nor two or five factor models fit

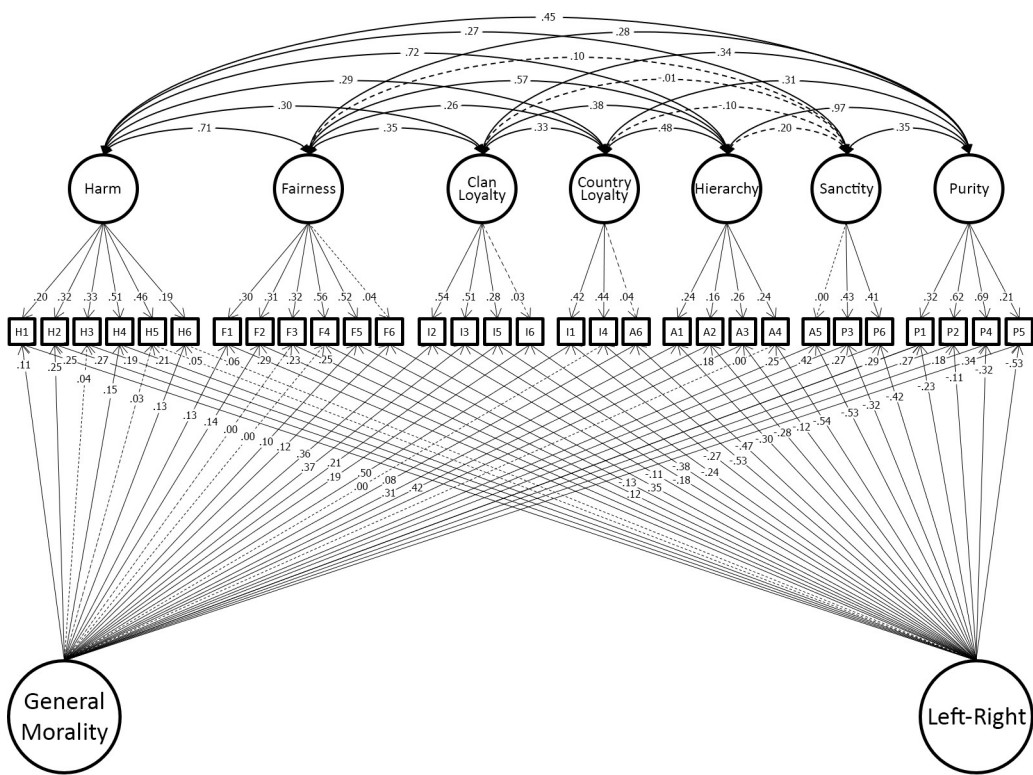

**Fig 4. Study 1 best-fitting model showing only the seven moral foundations and two general-factors (binding/individualizing and method variance paths in Fig 5 for clarity).**

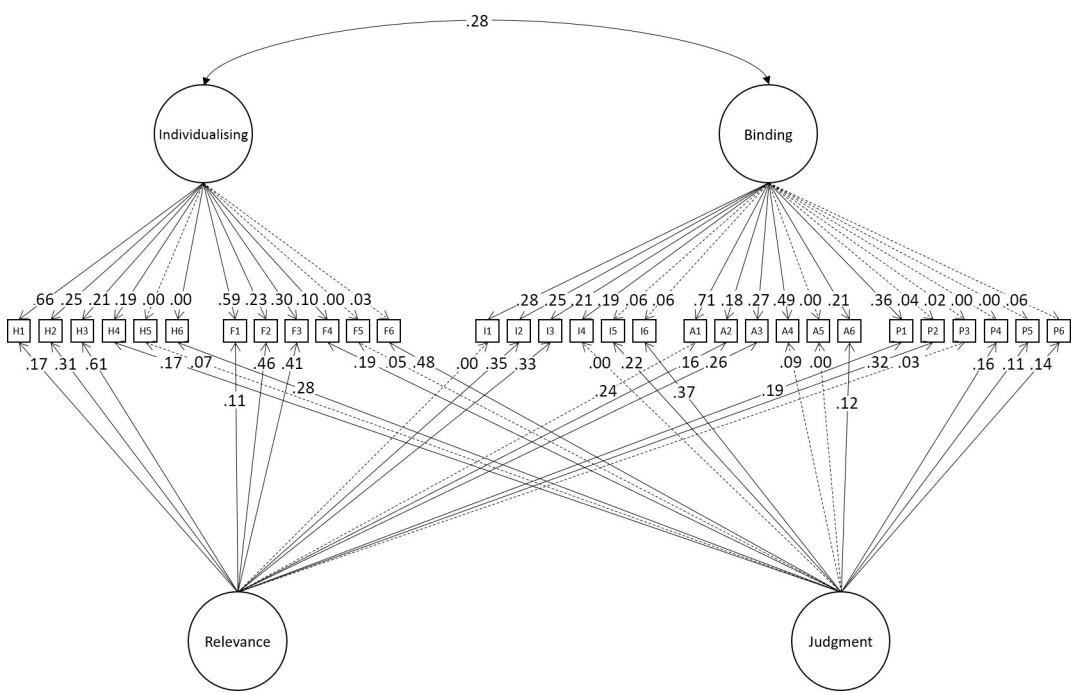

**Fig 5. Study 1 best-fitting model showing only the binding/individualizing and method variance paths (seven moral foundations and two general-factors in Fig 4 for clarity).**

was acceptable (though a five-factor model's fit was better compared to other models in this simple class). Building upon these findings, we were able to generate a well-fitting model by utilising Multitrait-Multimethod approaches, adding group and general factors and two additional moral foundations. The final model was considerably improved compared to previous analyses and achieved acceptable fit for both RMSEA and CFI metrics. Second, this well-fitting model replicated in a holdout data set. In studies 2–5 we will test replication in more depth, but this initial replication suggests that the model uncovered a reliable basic structure, though, this will require validation in a range of samples and cultures.

The third finding is perhaps most important and highlights theoretical implications for understanding moral foundations. The harm/care and fairness/reciprocity foundations reproduced with perfect fidelity: that is for each of these foundations, all 6 items loaded on a single factor in the 7-factor model supporting the MFT. By contrast, the well-fitting model draws firm distinctions between sanctity and purity and between loyalty to country and loyalty to what we termed clan (combining family and community). This seven (rather than five) foundation model broke-out items from the sanctity/purity, authority/respect, and ingroup/loyalty foundations to form independent sanctity and purity foundations, and independent foundations of loyalty to country and loyalty to clan. These changes also altered the nature of the authority foundation, leaving it more obviously aligned around hierarchy. The differences between 5-factor and our 7-factor models are depicted in Fig 6. We return to discuss these changes in more detail after testing replicability of the model in studies 2 through 4 below.

We also found that binding and individualizing factors were required, reflecting the covariation of item-responses driven by these two larger groupings. This supports a prediction of MFT and, again, we discuss this in more detail in the final discussion.

Finally, the model also required two general factors. One factor distinguishing the "tilt" or correlated changes in multiple foundations as one shifts along the liberal to conservative of left

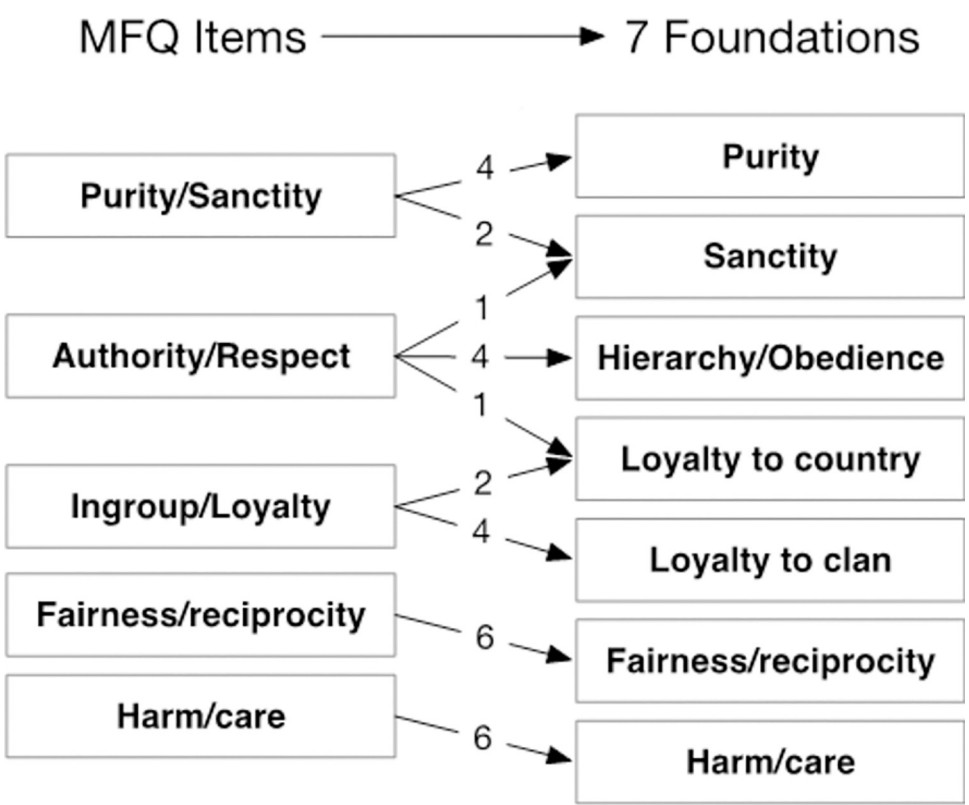

**Fig 6. Movement of items from the five MFQ foundations to the well-fitting 7 factor model.**

to right spectrum. This factor accounts for the otherwise inexplicable patterning of group and individual moral sensitivity along this dimension. Finally, a second general/acquiescence factor capturing tendency to overall higher or lower moral concern across the board, highlighting the need to include the dimension of moral vs. amoral in theorising on the moral foundations.

These findings are discussed in more detail in the general discussion after we report additional replications. The results in the holdout dataset closely replicated training dataset modelling: each step we took to improve the model in the training dataset also led to the improvement in the holdout dataset. The fit metrics in the holdout dataset, were comparable to those in the training dataset. The model thus met our first criteria for replicability: comparable fit in an independent holdout dataset. In studies 2 through 5, we test whether our best fitting model replicates in four large independent data samples.

## Study 2

In each of studies 2–5 we used a new independent sample, testing the fit of the eight models tested or developed in study 1, including the critical final, well-fitting model from study 1:

### Participants, measures and procedure

We used data from the 7,130 participants (2,815 females, 4,315 males; age M = 37.55, SD = 14.53) who participated in the Graham et al. study 3 [13]. Participants in this sample were adults, mostly from Western countries, who filled in the MFQ-30 questionnaire at www. yourmorals.org.

**Table 5. Model fit comparisons for the replication dataset in Study 2.**

| Model | EP | CFI | TLI | RMSEA | AIC | Compare with Model |
|---|---|---|---|---|---|---|
| M1. 2-Factor model | 91 | .750 | .731 | .078 | 252833.38 | Model 8 |
| M2. Hierarchical model | 101 | .809 | .789 | .070 | 248690.42 | Model 8 |
| M3. 5-Factor model | 100 | .813 | .794 | .069 | 248393.99 | Model 8 |
| M4. M3 + Method factors | 130 | .877 | .854 | .058 | 243862.97 | Model 8 |
| M5. M4 + Binding and individualising | 161 | .913 | .886 | .051 | 241367.97 | Model 8 |
| M6. M5 + Two new foundations | 172 | .931 | .906 | .046 | 240119.36 | Model 8 |
| M7. M6 + One general factor | 202 | .955 | .934 | .039 | 238379.91 | Model 8 |
| **M8. M6 + Second general factor** | **232** | **.971** | **.951** | **.033** | **237332.63** | |

AIC = Akaike information criteria; Best fitting model is printed in bold.

## Results

We constructed the models from Study 1 in the new data for Study 3 and examined their fit, with no additional modifications made. The best-fitting model from Study 1 was also best-fitting model in this dataset. See Table 5 for the model comparisons and fit statistics. The fit of the Model 8 (our best model in the Study 1) was satisfactory according to all three fit-metrics in this sample. Fig 7 shows the parameter estimations of the model's structural part in this dataset.

## Study 3

The age and sex composition in this sample is similar to that of Study 2. However, all participants in the Study 3 were either US citizens or US permanent residents whereas participation

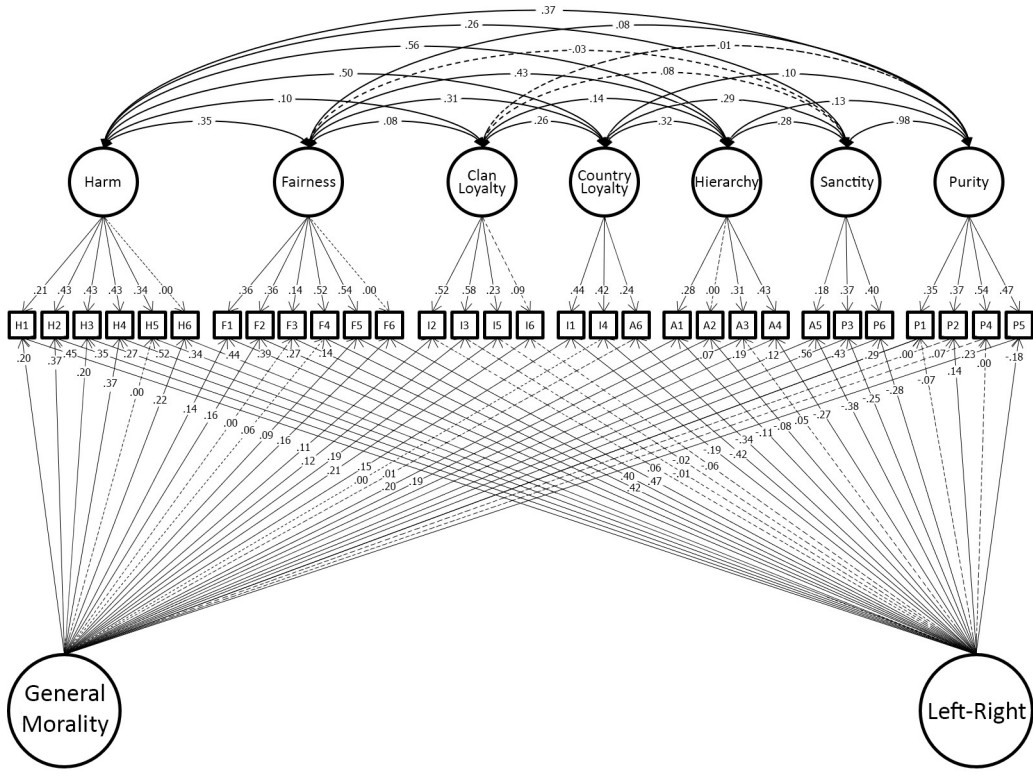

**Fig 7. Structural part of the final model in Study 2.**

in the Study 2 was open to everyone around the world. Even though most participants in Study 2 were still from the Western countries, US is markedly different in some personality traits that may be relevant to the moral judgment (e.g. higher religiosity, individualism [49]). It is therefore interesting to investigate whether the sample restriction to one country will affect the fit of the models developed in the Study 1.

### Participants, measures and procedure

We used data from the 1,052 participants (566 females, 466 males; age M = 39.66, SD = 12.46) who participated in the Smith et al. study [25]. The data were collected on the MTurk platform. To increase reliability of responses, only participants who had at least 99% Human Intelligence Task approval rate on the MTurk platform could participate.

### Results

We constructed the models from Study 1 in the new data for Study 3 and examined their fit, with no additional modifications made. The best-fitting model from the Study 1 was also best-fitting model in this dataset. See Table 6 for the model comparisons and fit statistics. The fit of the best-fitting model, model 8, was satisfactory according to RMSEA and CFI metrics, but not TLI. Fig 8 shows the parameter estimations of the model's structural part in this dataset.

## Study 4

The sample used in this study is comparable to that used in Study 3. Data were collected on the MTurk platform and all participants were from the US. Both studies also have similar age and sex composition. However, the data used in this study were collected in June-July 2018, whereas the data used in study 3 were collected in October 2014. This spread in time of almost 4 years spans some significant US events, including a contentious presidential election which potentially impact perception and responding to questions about moral issues. This sample, then, provides a useful test of the resilience of the new model to such changes.

### Participants, measures and procedure

We used data from the 591 participants (267 females, 284 males; age M = 39.48, SD = 10.62) who participated in the O'Grady et al., Study 2 [42]. Participants were hired on the MTurk platform. To increase reliability of responses, only participants meeting the "Master" or expert qualification on the MTurk platform suggesting that they are reliable and experienced workers were allowed to participate.

**Table 6. Model fit comparisons for the replication dataset in the Study 3.**

| Model | EP | CFI | TLI | RMSEA | AIC | Compare with Model |
|---|---|---|---|---|---|---|
| M1. 2-Factor model | 91 | .746 | .727 | .082 | 44888.66 | Model 8 |
| M2. Hierarchical model | 101 | .812 | .793 | .072 | 43950.13 | Model 8 |
| M3. 5-Factor model | 100 | .816 | .798 | .071 | 43887.89 | Model 8 |
| M4. M3 + Method factors | 130 | .875 | .851 | .061 | 43063.85 | Model 8 |
| M5. M4 + Binding and individualising | 161 | .910 | .882 | .054 | 42603.66 | Model 8 |
| M6. M5 + Two additional foundations | 172 | .929 | .904 | .049 | 42358.87 | Model 8 |
| M7. M6 + One general factor | 202 | .952 | .929 | .042 | 42017.3 | Model 8 |
| M8. M6 + Second general factor | 232 | .964 | .941 | .038 | 41917.68 | |

AIC = Akaike information criteria; Best fitting model is printed in bold.

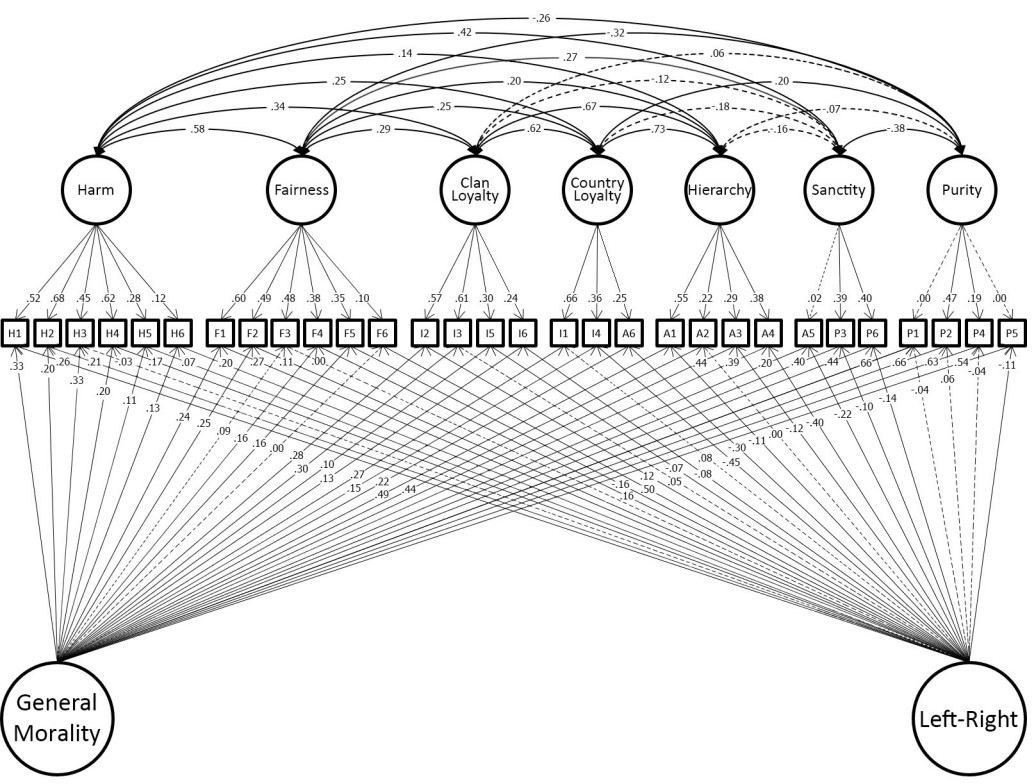

**Fig 8. Structural part of the final model in Study 3.**

## Results

We constructed the models from Study 1 in the new data for Study 3 and examined their fit, with no additional modifications made. The best-fitting model from the Study 1 was also best-fitting model in all four datasets in the Study 4. See Table 7 for the model comparisons and fit statistics. The fit of the best-fitting model, model 8, was satisfactory according to RMSEA and CFI metrics, but not TLI. Fig 9 shows the parameter estimations of the model's structural part in this dataset.

## Study 5

The sample used in study five differs in two regards from those used in studies 1–4. First, it is a non-Western sample (from China). Second, participants in this sample were significantly

**Table 7. Model fit comparisons for the replication dataset in the Study 4.**

| Model | EP | CFI | TLI | RMSEA | AIC | Compare with Model |
|---|---|---|---|---|---|---|
| M1. 2-Factor model | 91 | .698 | .674 | .103 | 19433.52 | Model 8 |
| M2. Hierarchical model | 101 | .793 | .771 | .087 | 18692.51 | Model 8 |
| M3. 5-Factor model | 100 | .801 | .781 | .085 | 18629.21 | Model 8 |
| M4. M3 + Method factors | 130 | .871 | .846 | .071 | 18105.93 | Model 8 |
| M5. M4 + Binding and individualising | 161 | .912 | .886 | .061 | 17808.67 | Model 8 |
| M6. M5 + Two additional foundations | 172 | .932 | .908 | .055 | 17667 | Model 8 |
| M7. M6 + One general factor | 202 | .964 | .946 | .042 | 17445.42 | Model 8 |
| M8. M6 + Second general factor | **232** | **.969** | **.948** | **.041** | **17436.24** | |

AIC = Akaike information criteria; Best fitting model is printed in bold.

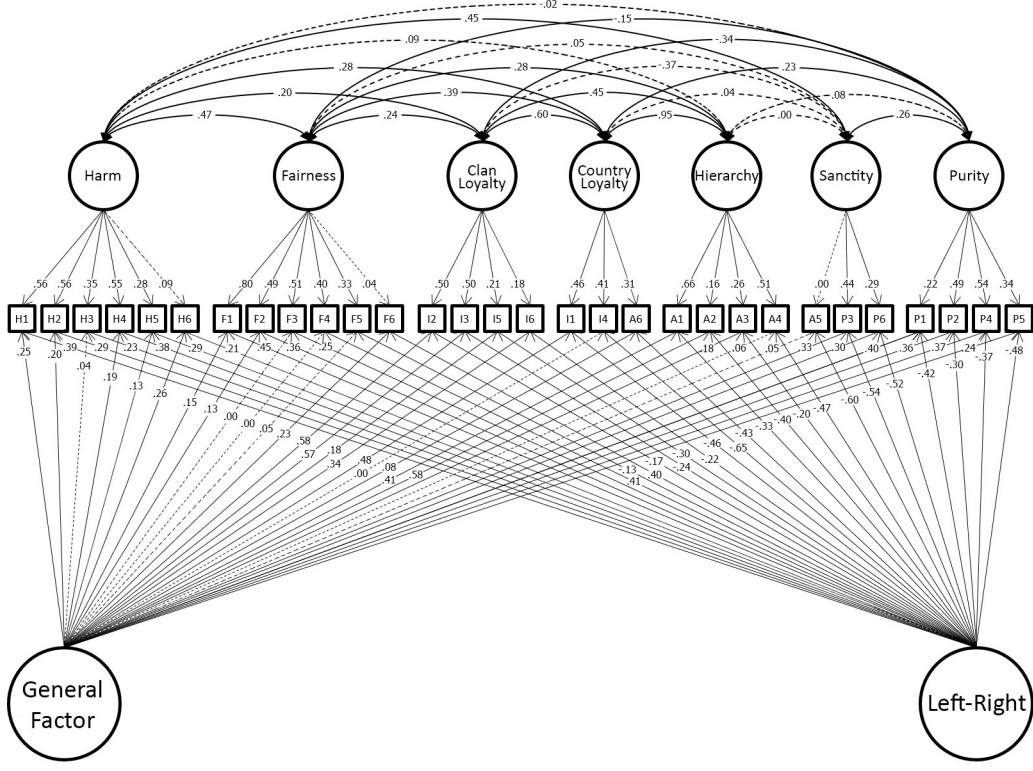

**Fig 9. Structural part of the final model in Study 4.**

younger than participants in the other samples used in this paper. While relatively small, this sample, then, offers an opportunity to investigate whether our model developed in a Western sample fits well in a sample that differs in both age and culture.

## Participants, measures and procedure

We used data from the 452 participants (355 females, 97 males; age M = 19.70, SD = 1.34) who participated in the Wang et al. study [43]. Participants were students from two Chinese universities. Moral foundations questionnaire used in this study was translated to the Chinese.

## Results

We constructed the models from Study 1 in the new data for Study 3 and examined their fit, with no additional modifications made. The best-fitting model from the Study 1 was also best-fitting model in all of the four datasets in the Study 5. See Table 8 for the model comparisons and fit statistics. The fit of the best-fitting model, model 8, was satisfactory according to RMSEA, but not CFI and TLI metrics. Fig 10 shows the parameter estimations of the model's structural part in this dataset.

## General discussion

The aim of the paper was to construct a well-fitting model of the MFQ, thus identifying accurately the structure of moral foundations, building and replicating the model over five studies. This final model preserved the fairness/reciprocity and harm/care foundations intact.

**Table 8. Model fit comparisons for the replication dataset in the Study 5.**

| Model | EP | CFI | TLI | RMSEA | AIC | Compare with Model |
|---|---|---|---|---|---|---|
| M1. 2-Factor model | 91 | .585 | .554 | .105 | 14639.74 | Model 8 |
| M2. Hierarchical model | 101 | .599 | .558 | .104 | 14582.19 | Model 8 |
| M3. 5-Factor model | 100 | .607 | .567 | .103 | 14546.8 | Model 8 |
| M4. M3 + Method factors | 130 | .794 | .754 | .078 | 13676.04 | Model 8 |
| M5. M4 + Binding and individualising | 161 | .835 | .785 | .073 | 13511.43 | Model 8 |
| M6. M5 + Two additional foundations | 172 | .871 | .826 | .065 | 13348.56 | Model 8 |
| M7. M6 + One general factor | 202 | .911 | .868 | .057 | 13183.86 | Model 8 |
| M8. M6 + Second general factor | 232 | .932 | .887 | .053 | 13115.09 | |

AIC = Akaike information criteria; Best fitting model is printed in bold.

However, the binding foundations divided into five, rather than three foundations. Purity/sanctity split into independent foundations of purity and sanctity; Loyalty/group divided into independent factors of loyalty to clan and loyalty to country. Finally, Authority/respect was refocussed on hierarchy, losing one item to the new sanctity foundation and another into loyalty to country. In addition to these 7 foundations, higher-level factors of binding and individualizing were supported, along with a general/acquiescence factor. Finally, a "moral tilt" factor corresponding to coordinated left-leaning vs. right-leaning moral patterns was supported. This new model of the Moral Foundations Questionnaire has several implications for moral foundations theory and suggests additional directions for research. Each of these is discussed below.

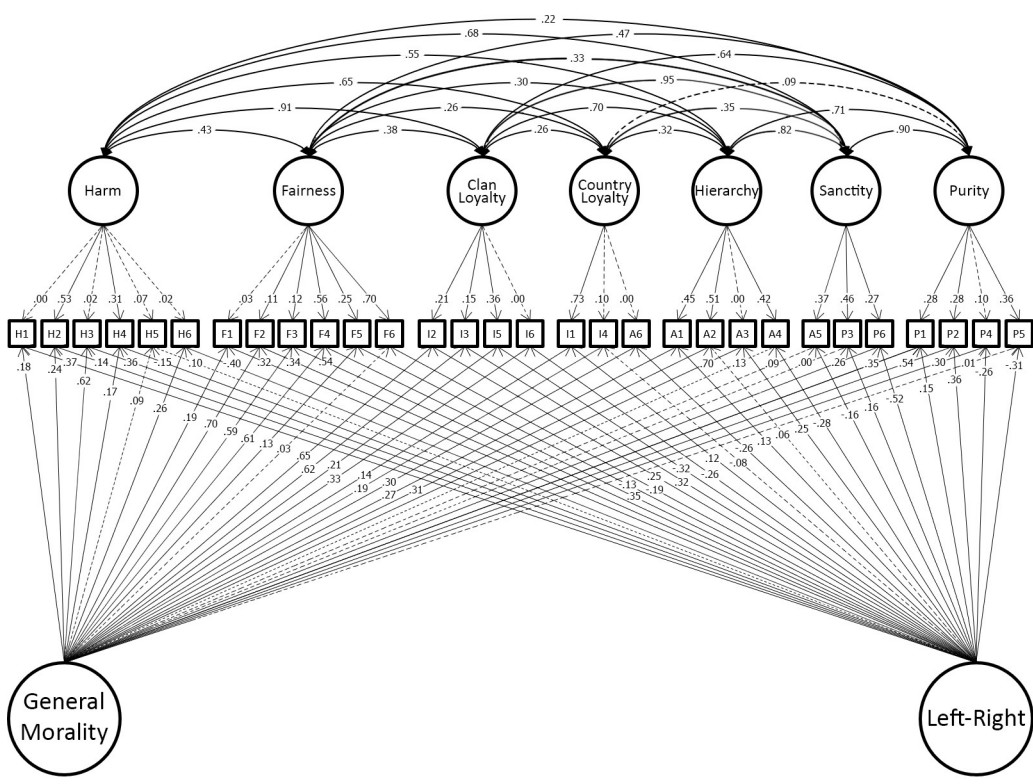

**Fig 10. Structural part of the final model in Study 5.**

## New moral foundations

In our modelling, two additional moral foundations, sanctity and loyalty to country were added, splitting off items from the sanctity/purity, authority/respect, and ingroup/loyalty foundations. To use Durkheim's [11] insight, sanctity focussed on "*sacred things… things set apart and forbidden*". This factor involved acting in ways that would be approved by God, including being chaste and performing roles assigned to us in society. This combining of attitudes towards religion with restrictive reproductive morals has been reported in practice reliably and across cultures [50]. By contrast, purity loaded on items involving moral support for the avoidance of disgusting and unnatural things.

This division of sanctity and purity is clearly recognised in the hyphenated name of the original, and our model formalises this division, allowing for distinct sensitivity to avoidance of disgusting or unnatural things: the "*guardian of the body*" [2] versus sanctification of "*things set apart and forbidden*" [11]. Individuals can independently respond weakly or strongly to these two distinct moral patterns, e.g. reacting strongly "*to elicitors that are biologically or culturally linked to disease transmission*" [2] while being less reactive to ritual and the sanctification of social structure. It will be of value in future studies to identify how this distinction sheds light on external phenomena such as complex multi-dimensional models of religiosity, in which elements of avoidance of vice and pursuit of virtue linked to sanctity might more tightly link to religiosity and emphases on purity may relate to distinctions among religions. Additional research might model how the distinction made here between sanctity and purity maps onto external measures such as complex multi-dimensional indices of religiosity [51].

The second novel foundation, which we identified as "loyalty to country", absorbed items from the ingroup/loyalty and authority/respect foundations. It distinguished people who are proud of their country, showing love and pride in their nation's history as well as duty to country from those who are not, including not picking and choosing which orders from leaders they would follow, but obeying out of a sense of duty to country despite disagreement with a specific policy. Patriotism or nationalism has previously been identified as a unique and important foundation by Haidt [10], contrasting this with globalism to create a dimension of caring for the people in one's own country more than those of other countries, versus treating all people as identical in terms of their moral call upon us.

The "loyalty to clan" foundation emerged from what remained of the original ingroup/loyalty foundation. Having lost one item to the new sanctity foundation and another migrating to loyalty-to-country this foundation was refocused almost exclusively around loyalty to family and immediate group. Such preference for kin is predicted from kin selection theory [52] and reciprocal altruism [53]. By contrast, in modern diverse and large-scale societies loyalty to country implies altruism towards non-kin, a different and evolutionary novel mechanism. Future research should test whether the distinction between loyalty to clan and loyalty co country emerges in small monoethnic countries.

Finally, the foundation we termed Hierarchy emerged from the original authority/respect foundation, refocused tightly on respect for authority and tradition and preference for order over disorder and chaos. This preference for order and obedience to hierarchy was thus rendered distinct from loyalty per-se, which was now moved to separate foundations. This distinction of loyalty to one's group and obedience to hierarchy, has a distinguished history in theory. In his work on administrative behavior, Simon [54] identified loyalty and obedience as the two necessary conditions for the existence of organisations, defining loyalty as the capacity to introject organizational objectives in place of one's own aims and obedience as choosing to make one's default response be to follow requests of a superior: a definition which corresponds closely to notions of respecting the wishes of those in authority. The new model formalises

these distinct aspects of moral concern which are merged in the 5-domain model of the MFQ, distinguishing concern for one's countrymen (loyalty to country) concern for family and kin, and concern for the organization and obedience (authority).

A future theory of moral judgment will need to propose and test hypotheses about the selective pressures on loyalty to country vs loyalty to kin and purity vs sanctity factors. In addition, constructing a modified measure of moral foundations may require generating new items to cover the larger number of domains.

## Group factors

An early change which improved model fit greatly was the inclusion of group factors representing binding and individualising foundations. Rather than being implemented hierarchically, as has been done previously, these group factors were implemented as a bi-factor structure, at the item level. The extra degrees of freedom provided by a bi-factor model relative to a comparable hierarchical model can, in exploratory cases, be led to model misspecification by modelling sample-specific variance. It is recommended, therefore, that bi-factor models should be validated in new data sets [55]. We did this, testing replication of the model in four independent datasets. Two points are worth discussion regarding these group factors. First, the group factors strongly confirm a predicted component of the moral foundation theory. They highlight the distinct role played in moral perception by the units of the individual, and of the group, organizing moral concerns separately around people and groups, and preserving these levels of value across related foundations. This highlights the second note-worthy aspect of modelling, namely that these group factors fit best when implemented as impacting the items directly rather than working via higher-level latent constructs. This suggests that much as specialised aspects of visual world are processed by regions specialised for colour or motion, concern for the individual and concern for the group may themselves be processed as distinct mechanisms in the "moral mind", recognising tagging and recruiting behavior across domains. These factors warrant additional study.

## General factors

Adding an unconstrained general factor to the model significantly improved fit. This factor instantiates a moral dimension linked to views across all the moral domains, organized in a mono-dimensional manner. This factor loaded positively on fairness and harm items–especially items emphasizing compassion and equality–and negatively on items related to authority and on purity. Higher scorers were both more somewhat more likely to respond positively to items such as "*I think it's morally wrong that rich children inherit a lot of money while poor children inherit nothing*" and much less likely to agree that "*Men and women have different roles to play in society*" or to be "*proud of my country's history*". This pattern of loadings corresponds to a dimension of liberal-conservative or left-wing/right-wing views. As such, it can explain the difference in moral foundations identified by Haidt and Graham [2], moving from liberal egalitarianism to conservatism, including the subtle patterning of this move. Future study could usefully focus on understanding this factor, including relating it to other constructs, for instance one or other component of the social dominance construct [48], which was previously offered as an explanation of liberal-conservative differences [56]. Such an alignment could also arise from several causes, for instance philosophical differences, or unmeasured factors such as socio-economic self-interest factors which might align views and interests across multiple domains. Additional studies will be required to investigate the basis of this factor.

The second general factor was constrained to load positively on all items. As such it functions as a general morality factor, loading on all moral attitudes and varying coherently from

the low degree of concern for any of harm/care, fairness/reciprocity, ingroup/loyalty, authority/respect, purity/sanctity to high levels of regard for all these foundations. One candidate for explaining such a factor with low concern for morality at the low-end would be psychopathy [57]. This dimension maps onto reckless (unconscientious, now-focussed, goal-lessness) and disagreeable (angered if not getting one's way) personality and may reflect effects of personality. Making these possibilities testable, for instance by using measures of psychopathy (e.g. [58]) is a benefit becomes straightforward given the new model. While such an interpretation of this factor as representing a genuine "general factor" of moral judgment is possible, other explanations remain plausible, in particular factors related to modelling bias and acquiescence, and we turn too these important but more technical matters in the next section.

## Modeling implications

There are at least two reasons why good fit of the model is important. From a theoretical standpoint, good fit strengthens confidence that the elements of the model represent true relationships. From a practical perspective, well-fitting model can be expected to provide better predictions, just as a better-focused lens gives clearer vision. The MFQ incorporates two methods– relevance and judgment–to measure each foundation. Surprisingly, these are rarely modelled explicitly in the moral foundations' literature and our modelling clearly shows that treating these item-types as distinct and modelling this method variance significantly increased the fit of the model. This increase in fit is in-line with Curry et al. [24] who also found that modeling measurement effects improved fit. Aspects of model design also impact improvements in fit when modeling the group factors of binding and individualising. These clusters were proposed by the MFT creators [13], but previous attempts to incorporate group-factors focussed on hierarchical implementations, where variance from the group factors must pass through the 5-factor structure. These, as covered in the introduction, failed. By contrast, we modelled these group-factors at the item level and this improved model fit significantly. Similar improvements from bi-factor modeling have been reported for cognitive ability models [59]. This methodological move to a bi-factor implementation also suggests something about the mechanism of the binding and individualising factors: namely that the variance they capture does not work via the foundational domains but rather coordinates behavior directly.

The second general factor was constrained to load positively on all items. As noted above, this may represent a substantive general "morality" factor. Researchers are increasingly aware of the effects of acquiescence bias–the tendency of respondents to agree (or disagree) with all questionnaire items, regardless of their content. Acquiescence bias could be eliminated from the questionnaire by including reverse-coding items but the MFQ does not contain such items. A useful research project would be to test this using new high-performance reverse coded items or measuring acquiescence and testing if the general factor is related to this. We must also suspect social desirability [40] as an influence on any measure linked to socially-evaluated outcomes. Social desirability emerges when subjects over-report desirable traits due either to a cognitive bias or conscious impression management [60]. Future studies should investigate this possibility, for instance by including a self-perception bias measure (e.g. [61]) and testing evidence for a relationship between the general factor of morality and bias tagged in this way. Constructing a better measure of moral judgment may also require rephrasing of some items and perhaps creating new ones to remove or at least minimise the effects of method variance and possible bias [62]. Another important opportunity for future research is external validation of the seven moral judgment factors. Here we used multiple datasets to establish that seven factors emerge even in demographically diverse samples. Future research should complement this by testing the ability of the new model to predict relevant outcomes

such as social and political attitudes, in particular, political affiliation [13, 25], willingness to donate to charitable causes [63], attitudes towards religion [20] and one's country patriotic symbols [19] with the prediction that scores from this 7-factor model will explain such outcomes better than five-factor scores can. Additional opportunities for development posed by the better-fitting model include the opportunity to generate additional items so that each of the facets of the seven-foundation model are equally well represented.

## Limitations

It is important to note that while our final model was the best-fitting model in all samples, the precise path estimations differ between the samples. Such variation, however, is expected by chance. A strength of the paper is its use of multiple samples and not all from the same culture though, given that we used only one non-Western sample, this requires validation. In study 2–5, we extended the findings of study 1 by testing our model in four independent replication datasets including one non-Western sample. In all of them, our model performed better than any other model previously reported. In the first and largest replication dataset (Study 2), all three fit metrics we used (TLI, CFI and RMSEA) were satisfactory. In the second and third datasets (Studies 3 and 4) only RMSEA and CFI were satisfactory whereas in the fourth dataset (Study 5) only RMSEA was acceptable, which can be explained by the non-Western nature of the sample. Lower fit in the non-Western sample may result from cultural differences, alternatively, some concepts or phrasings in the MFQ may not have exact counterparts in all languages. This sample also had higher proportion of females (78%) compared to other samples. It also showed the largest model fit improvement by modelling two method factors. To distinguish between these possibilities, future studies may investigate whether responses to the English and foreign language versions of MFQ differ in bilingual samples from non-Western populations. It is also worth mentioning that in Study 2 we used data provided by participants who voluntarily visited a moral psychology website and, presumably, were highly motivated. Similarly, in Study 3–4 we used data provided only by MTurk participants with high approval rate. This may have affected how representative of general population these samples are. Notably, however, in all four replication samples the rank order of the models tested was the same, including the non-Western one. This suggests that our model is robust to replication across sample characteristics.

## Summary

The five studies reported here involved five important changes in modeling of the moral foundations questionnaire and yielded a substantially improved, well-fitting and substantively distinct model of moral foundations.

Of theoretical relevance, at the foundational level, two additional foundations were needed. The first formally implemented the distinction between sanctity and purity domains of purity/sanctity foundation recognised in Durkheim's "sacred things" [11] and what Haidt and Graham [2] described as "guardian of the body". The second involved distinguishing a foundation of loyalty to country in addition to foundations of loyalty to clan consisting of four ingroup items. This, we suggested maps onto important theorised roles of a moral dimension currently best displayed in support for the nation versus globalisation [10]. The distinct foundations of loyalty and respect with their critical functions in allowing people to operate within organizations [54] were retained, but no longer overloaded with national-level moral concern.

A third theoretically important change, the necessity of formal "individualizing" and "binding" factors were required, reflecting strong association among the foundations linked to compassion for the individual, and among domains concerning hierarchy, social norms, and

survival of the group. The model also required a general factor implementing loading with opposite sign on the binding and individualizing foundations. This theoretical novelty account for the otherwise inexplicable left-right dimension which "tilts" the moral foundations in a coordinated fashion [13]. Finally, a general/acquiescence morality factor loading positively on all domains was required for a good fit.

A well-fitting model of personality should reveal factors which are important both theoretically and which, more closely reflecting the causal structure, should better-predict relevant outcomes compared to existing models. The MFQ can be scored using the 7-factor model presented here, together with group factors and general effects to test its predictive validity in past and future studies in comparison to the classic two-cluster and five-foundation scoring systems. The new model may increase the variance MFT can account for in domains such as politics [13, 17], differences among religions in emphasis on ritual versus purity [20], studies of organizational behavior requiring institutional loyalty and respect [54], new work on patriotism and links to constructs such as globalization and the need for social or trait accounts for this domain, as well as studies linking generally low scores on the MFQ to research on psychopathy.

## Author Contributions

**Conceptualization:** Michael Zakharin, Timothy C. Bates.

**Project administration:** Timothy C. Bates.

**Software:** Timothy C. Bates.

**Supervision:** Timothy C. Bates.

**Writing – original draft:** Michael Zakharin.

**Writing – review & editing:** Timothy C. Bates.

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
