## [Decision Letter · Decision Letter 0]

22 Jul 2021

PONE-D-21-17063

Remapping the foundations of morality: Well-fitting structural model of the Moral Foundations Questionnaire.

PLOS ONE

Dear Dr. Zakharin,

Thank you for submitting your manuscript to PLOS ONE. After careful consideration, we feel that it has merit but does not fully meet PLOS ONE’s publication criteria as it currently stands. Therefore, we invite you to submit a revised version of the manuscript that addresses the points raised during the review process. Please submit your revised manuscript by Sep 05 2021 11:59PM. If you will need more time than this to complete your revisions, please reply to this message or contact the journal office at plosone@plos.org. Please include the following items when submitting your revised manuscript:

A rebuttal letter that responds to each point raised by the academic editor and reviewer(s). You should upload this letter as a separate file labeled 'Response to Reviewers'.A marked-up copy of your manuscript that highlights changes made to the original version. You should upload this as a separate file labeled 'Revised Manuscript with Track Changes'.An unmarked version of your revised paper without tracked changes. You should upload this as a separate file labeled 'Manuscript'

We look forward to receiving your revised manuscript.

Kind regards,

Peter Karl Jonason

Academic Editor

PLOS ONE

Journal Requirements:

Reviewers' comments:

Reviewer's Responses to Questions

**Comments to the Author**

1. Is the manuscript technically sound, and do the data support the conclusions?

Reviewer #1: Yes

Reviewer #2: Yes

2. Has the statistical analysis been performed appropriately and rigorously? 

Reviewer #1: Yes

Reviewer #2: Yes

3. Have the authors made all data underlying the findings in their manuscript fully available?

Reviewer #1: No

Reviewer #2: Yes

4. Is the manuscript presented in an intelligible fashion and written in standard English?

Reviewer #1: Yes

Reviewer #2: Yes

5. Review Comments to the Author

Reviewer #1: Review of “Remapping the foundations of morality”.

After review, I think the paper requires revision. Plos offers minor or major revision, I don’t know what those mean, as an editor I stick with accept, revise or reject.

The strength of the paper is its empirical analyses, but it requires substantial re-framing and engagement with the literature.

So, while I’m supportive of revision for certain, there’s some work to be done. I think it is appropriate to disclose my identity given I’ve contributed directly to this area, so as to be transparent for any potential COI (Pete Hatemi).

When Kevin Smith and I began looking into MFT, we found it both elegant and interesting, and started with the idea to provide evidence for genetic influences and one causal path for genetic influences on political attitudes. The data came out the other side; almost none of what Haidt and co proposed about the measures appeared to be empirically supported.

And here is where the current paper seems really out of touch with the literature, almost treating the question of the MFQ validity as a open question, when in fact, in dozens of studies, it has been shown to be invalid. So it is not true to say that “the predictive validity of the MFQ has been generally supported, especially in the domain of politics”. Rather just the opposite. What has been supported are correlations between the two, that’s about it.

So, let’s go back to the beginning. Kevin and I were perhaps among the first wave of people to find that the MFQ does not reliably factor into 5 dimensions in 10 populations (Smith et al- which btw your paper cites incorrectly), but at best 2, the structure differed by country (US and AU). This paper was not a critique and in fact we had no interest in finding a different factor structure. I suggest a reread of it. Then came Iurino and Saucier and many others who hit this more definitely (see Harper and Rhodes and Davis et al who find MFT is not valid across non US/white populations, and just the other day Hadarics Márton’s paper, then of course, the heterogeneity in MFTs within and across ideologies identified by Frimer; then finding that measure is not stable across time (2 years= Smith et al), and then the findings that MFT appears to be caused by, rather than cause, the traits it is proposed to influence (for attitudes or political orientation see Hatemi Crabtree and Smith , Kivikangas et al, Ciuk, Everett et al., Strupp-Levitsky et al (Jost), and Márton and Kende among many others . Finally it’s not heritable (Smith et al.).

In this, I think the paper needs to redo the front end. The above papers are not critiques, so it a bit disingenuous to frame them as such. Rather, most of them were studying a trait of interest, and in so doing found out the predictor variable (MFQ) was garbage.

If the current paper wants to restructure the MFQ then it has to do two things.

First it has to meaningful engage the some 20+ papers that can’t replicate the proposed factor structure. The comparison of fits in the paper seem highly selective.

So, a more clear and thorough review of MFQ/MFT’s serious shortcomings is needed to properly situate the paper and compare your factor structure with the 2, 5, 7, and other outcomes. Simply ignoring the works above, or attempting to frame them as critiques on the side is not good science in my view. It is not that a handful of scholar’s challenge MFT. It is that the most serious empirical explorations of the MFQ, with large samples and good data, find no evidence to support most of MFT’s claims, regarding the measure.

Doing this can be easily done. Read the papers, compare your approach and results, and place them in context.

The second thing, will take less leg work but a bit more thought. What the current paper proposes is that Haidts MFT is simply wrong. One cannot simply just restructure the MFQ into different domains, without then updating the theory. A read of Haidt’s book here is critical –The MFQ is the proposed measure of Haidt MFT theory. It has specific logic, organizing principles evolutionary roots etc that link the domains to each other and the MFQ questions. Certainly one can simply data drill, as Kevin and I did in the Human Nature paper without any theory. In this third paper Hatemi and Smith, we ran more EFT’s on the MFQ just to satisfy a reviewer to see if we could make the thing heritable, but that was in an SI. Here the paper is centered around offering a different factor structure to the data that conflicts with the theory as written. What are you competing hypotheses? What parts of your findings invalidate Haidt’s theory? What part support it ? What parts of Haidt’s theory require modification based on your results, what part require abandonment?

This is important because if one wanted to use this new formulation as a means to lets say, run different analyses, like behavior genetic ones for example, then this should be the paper to address those questions. Otherwise it is simply a data exercise – which is not a bad thing, but just limited in what it offers.

So, yes, I’d like to see this paper in print. My suggestions Read the lit, reframe the paper by engaging it more appropriately, mainly it is not that there are critiques, but rather Haidt’s theory and measure simply don’t hold up in a lot of the data. It is well established that the factor structure certainly is not supported. In most every large and nationally rep study its doesn’t work- though it seems to work with Haidt and Grahams student and internet samples. So, the question is what is the actual measure doing , what is the ideal factor structure and if it’s not what the theory proposes what does this say about the theory? Based on your proposed structure- is it still moral foundations?

One important and serious concern. You need permissions to use other peoples replication data for anything other than replication. Posting data for replication is for replication. Here you are using replication data for novel purposes. Since you used my data and I was never asked, I’m suspecting you did the same for others. It is an ethical question to take replication data in the manner you’re using it. Smart move here is to ask. It is very low cost , the price of an email and usually results in goodwill. If you dont, you risk your paper getting retracted. Not a smart play in my view. But risk is certainly a trait with individual differences and variation is genetically influenced at that.

Feel free to take my comments print them out and use them for TP, or to improve the paper as you see fit. Hopefully they help.

P

Reviewer #2: This is one of the empirically strongest papers on the Moral Foundations Questionnaire, with strengths that the paper itself accurately touts. But there are some caveats and weaknesses of which the authors seem unaware.

1. The samples are – except for Study 5, the smallest sample – based on predominantly Western respondents (in fact, predominantly Anglo-American). Page 16 claims that study 1 identified a ‘reliable basic structure’ but this might be true only for certain populations. In general, the paper needs more caveats about potential inapplicability to non-Western populations. I don’t think we should be having a few Chinese university attendees representing (standing in for) the entire non-Western world as is done here.

2. With respect to the one non-Western sample, the paper fails to note important details (the sample was evidently almost 80% female) and there are some questions. What universities were the respondents from and how Westernized were they? Why was this particular sample (among all non-Western samples administered the MFQ) chosen, and did that involve cherry-picking the sample most likely to be supportive of the select model?

3. One could also wonder if there is some degree of cherry-picked subjects in studies 1-4. The description of the study 1 sample does not rule out that these are all moral-psychology enthusiasts who volunteered. The study 2 was highly motivated – found their way to a website on moral psychology. Study 3 and 4 samples were from MTurk, but might be called ‘cream of the crop’ MTurkers based on the selection criteria; that’s well and good unless it means these results are dependent on using ‘elite questionnaire-response professionals’ and are ungeneralizable to a typical population. What percentage of MTurk participants meet the criteria specified, and were any MTurk participants recruited but then eliminated (in order to make favorable results more likely)? (A general take-home message might be that the intended structure of the MFQ is fragile, hard to locate in your typical noisy data.)

4. The models employed are entered in a particular order that was apparently set a priori (is there a pre-registration to verify this?), but the question is raised whether, when all is said and done, one (or more) of them might be unnecessary. That is, perhaps by backward removal of the element adding least, a more parsimonious outcome might be reached. This is analogous to the forward versus backward methods in stepwise regression. Related questions: How does it make sense to enter a hierarchical model (which includes the five-factor model) before one enters the five-factor model, and what is the difference between ‘two-factor model’ and ‘binding and individualizing’ foundations in, for example, Table 8.

5. Following that same thread: The conclusion (though unstated) would seem to be that one can use the MFQ, but must really analyze it or make sense of its scores in a complicated and onerous manner. That is perhaps a problem for construct validity. An important area of future implications would be this: What do these results suggest about how to construct a better moral ‘foundations’ measure (e.g., one that would not be so profoundly affected by method variance, or by the pull of an underlying two-factor model and of political leanings and of acquiescence or general morality vs. amorality? In other words, how can the measurement of moral foundations be cleaned up in these regards?

Smaller matters:

-The middle paragraph on page 26 leads to considerable head-scratching. The first sentence is incomprehensible. As for the 2nd, why is social dominance picked out so prominently among all possible alternatives? What is the ‘alignment’ referred to there? It would seem that one interpretation of the results is that ‘tilt’ and binding-individualizing make independent contributions because one cannot reduce the latter entirely to tilt (as some treatments of the topic have implied in the past), there being both liberal and conservative ways of endorsing binding AND of endorsing individualizing morality.

-In Table 8 it is noteworthy that moving from model 3 to model 4 yielded a huge improvement in fit, which was not the case in other samples. Does this say something about how Chinese respondents characteristically handle morality or this questionnaire?

-Page 13 identifies a set of three MFQ items and labels them as sanctity, but it is hard to apply this interpretation to the third item mentioned (about roles for men and women). One could just as easily label this factor as ‘traditional gender roles/expectations’ (that also would fit 2 of the 3 items).

-Page 13-14 separate out a patriotism factor from a loyalty factor, but it would seem more informative to label one of them as loyalty to country and the other as loyalty to family/team/group (i.e., to smaller-scope entities). In reference to same on page 24, it is implied that Herbert Simon differentiated patriotism from loyalty, but this seems unlikely given how Simon’s viewpoint is stated. Clarity needed.

-Until the very last part of the paper, there is a tendency to label one latent variable -- created by constraining all items to load positively on it – as General Morality although an equally plausible interpretation is Acquiescence. Would be better to mention both possible interpretations from the beginning.

-In Table 1, the “(modified)” notation for the best-fitting model for the Iurino and Saucier (2018) paper needs some explanation.

-The tendency to overestimate one’s personality traits is (on p. 12) mischaracterized as halo effect. Halo effect is more commonly used to refer to overestimation of someone else’s positive traits. Doing it to yourself is self-enhancement or social desirability bias.

-It seems that some text on pages 17-18 is redundant with what was said before in the paper, this needs a check.

6. PLOS authors have the option to publish the peer review history of their article (what does this mean?). If published, this will include your full peer review and any attached files.

Reviewer #1: No

Reviewer #2: No

---

## [Author Response · Author response to Decision Letter 0]

5 Sep 2021

Comments from Reviewer 1

Comment: The strength of the paper is its empirical analyses, but it requires substantial re-framing and engagement with the literature.

Response: We thank the reviewer for their positive evaluation of the empirical analyses and for their direction to additional citations in the literature. We have incorporated these below. 

Comment: And here is where the current paper seems really out of touch with the literature, almost treating the question of the MFQ validity as an open question, when in fact, in dozens of studies, it has been shown to be invalid. So it is not true to say that “the predictive validity of the MFQ has been generally supported, especially in the domain of politics”. Rather just the opposite. What has been supported are correlations between the two, that’s about it.

Response: Thank you for this suggestion. We were happy to reword this sentence removing the implication of prediction. We now say “Since its development, the correlation of the MFQ with external measures has been widely studied, especially in the domain of politics.” (See lines 118-119 page 5) 

Comment: So, let’s go back to the beginning. Kevin and I were perhaps among the first wave of people to find that the MFQ does not reliably factor into 5 dimensions in 10 populations (Smith et al- which btw your paper cites incorrectly), but at best 2, the structure differed by country (US and AU). This paper was not a critique and in fact we had no interest in finding a different factor structure. I suggest a reread of it.

Response: We apologise for not noting that Smith et al. (2017) and Hatemi et al. (2019) used multiple samples. Also, for using the Smith et al. (2017) reference to cover both Smith et al. (2017) and Hatemi et al. (2019) work. We have corrected the reference and now say (see line 136-140, pages 5-6)

“Smith et al. and Hatemi et al. [25, 26], using multiple samples, reported that the MFQ did not reliably factor into 5 dimensions, but rather into 2, with structure differing between the US and Australia. They also found that moral foundations are not stable across time and that MFQ scores reflect rather than cause political attitudes. Smith et al. [26] also reported that the MFQ foundations show no evidence of heritability” 

Comment: Then came Iurino and Saucier and many others who hit this more definitely (see Harper and Rhodes and Davis et al who find MFT is not valid across non US/white populations, and just the other day Hadarics Márton’s paper, then of course, the heterogeneity in MFTs within and across ideologies identified by Frimer; then finding that measure is not stable across time (2 years= Smith et al), and then the findings that MFT appears to be caused by, rather than cause, the traits it is proposed to influence (for attitudes or political orientation see Hatemi Crabtree and Smith, Kivikangas et al, Ciuk, Everett et al., Strupp-Levitsky et al (Jost), and Márton and Kende among many others . Finally it’s not heritable (Smith et al.).

Response: Thank you for this helpful comment: We now cite each of these papers mentioned in the revision. We say (See lines 141-151 page 6 and lines 185-191, page 9):

“Strupp-Levitsky et al. [27] suggest that rather than being causal, the foundations build on other, more basic, variables such as empathy, need for closure and need for cognition. In a study manipulating partisan and group identity cues by embedding these in modified MFQ items, Ciuk [28] reported that item endorsement was affected by partisan alignment, supporting the conclusion that causality runs from political ideology to moral foundations. At the psychometric level, Iurino and Saucier [5] tested measurement invariance of the 5-factor structure of the MFQ in 27 countries and concluded that there was little support for a five-factor solution for the questionnaire. Similarly, in a US sample, Davis et al. [29] tested measurement invariance of the MFQ in Black and White samples, concluding that the assumption of scalar invariance could not be supported. Jointly, these pose considerable challenges for a measure that aims to be culturally universal, perhaps especially problems in finding a well-fitting model as a basis for prediction.” 

We continue

“More recently, Harper & Rhodes [32] tested the factor structure of the MFQ in two British samples (total N = 750), confirming that the proposed five-factor structure was not psychometrically sound according to accepted metrics. They also tested an extended MFQ, including the nine items of the sixth “Liberty” foundation proposed by Haidt and colleagues [12]. Adding the Liberty scale, however, did not lead to a well-fitting six-factor model, and instead was better explained by a three-factor model comprising “traditionalism”, “compassion” and “liberty”. 

Comment: In this, I think the paper needs to redo the front end. The above papers are not critiques, so it a bit disingenuous to frame them as such. Rather, most of them were studying a trait of interest, and in so doing found out the predictor variable (MFQ) was garbage.

Response: We have clarified that these papers use the MFQ as a trait of interest. We say “moral foundations theory and the MFQ measure have been criticized both on theoretical and empirical grounds (e.g. by papers using it as a trait of interest)”. See line 127-129, page 5.

Comment: The comparison of fits in the paper seem highly selective.

Response: We are not quite sure what this comment refers to. We added two new references to Table 1 (see page 8), citing Harper & Rhodes (2021) and Hadarics & Kende (2017) findings, both supporting the conclusion that MFQ falls short of the acceptable degree of model fit. In the text we now say “No reports, to our knowledge, have resulted in satisfactory fit (see Table 1 for a sample of fits for different MFQ models in different samples and cultures).”, lines 168-170, page 7.

Comment: …One cannot simply just restructure the MFQ into different domains, without then updating the theory. …The MFQ is the proposed measure of Haidt MFT theory. It has specific logic, organizing principles evolutionary roots etc that link the domains to each other and the MFQ questions. [The present] paper is centered around offering a different factor structure to the data that conflicts with the theory as written. What are you competing hypotheses? What parts of your findings invalidate Haidt’s theory? What part support it? What parts of Haidt’s theory require modification based on your results, what part require abandonment?

This is important because if one wanted to use this new formulation as a means to lets say, run different analyses, like behavior genetic ones for example, then this should be the paper to address those questions. Otherwise it is simply a data exercise – which is not a bad thing, but just limited in what it offers.

Response: We agree that enumerating which parts of our findings invalidate Haidt’s theory and which support it, and, thus, which parts of Haidt’s theory require modification based on our results is important, and we strove in the discussion to do this. We have taken a second look at the discussion rewriting where possible to make the differences and similarities between the MFQ predictions and our seven-factor model clearer. Regarding the individualizing foundations we now say on lines 416-425, page 18: 

“The harm/care and fairness/reciprocity foundations reproduced with perfect fidelity: that is for each of these foundations, all 6 items loaded on a single factor in the 7-factor model, supporting the MFT.”

Regarding three binding foundations we say: 

“By contrast, the well-fitting model draws firm distinctions between sanctity and purity and between loyalty to country and loyalty to what we termed clan (combining family and community)”. 

We have also added a figure showing the mapping from the 5 hyphenated original foundations to the 7-factor model. (Figure 6).

This suggests a need to modify MFT to account for additional distinct evolutionary selection pressures which would cause distinct sanctity and purity systems (rather than a single system processing both these kinds of information) and distinct systems for loyalty to kin and to country (rather than a single system processing both these kinds of information).

In this work we were constrained by the existing MFQ items which limits our ability to propose and test competing hypotheses. We did not enter with competing hypotheses, but rather generated these in interpreting the results of study 1, before replicating them in four independent datasets. We plan to test our model in future work and regarding this we now say (on lines 609-612, page 26):

“A future theory of moral judgment will need to propose and test hypotheses about the selective pressures on loyalty to country vs loyalty to kin and purity vs sanctity factors. In addition, constructing a modified measure of moral foundations may require generating new items to cover the larger number of domains”

Comment: One important and serious concern. You need permissions to use other peoples replication data for anything other than replication. Posting data for replication is for replication. Here you are using replication data for novel purposes. Since you used my data and I was never asked, I’m suspecting you did the same for others. It is an ethical question to take replication data in the manner you’re using it.

Response: Thank you for noting this. We have emailed the corresponding authors informing them about our use of their data in our work, receiving positive replies from all of them. We note in the text the licensing of each of these open access datasets (Studies 2-5) covered by licenses allowing any use of data, not only replication, as follows (see lines 766-775, pages 32-33):

Study 2: CC0 license (You can copy, modify, distribute and perform the work, even for commercial purposes, all without asking permission)

https://dataverse.harvard.edu/dataset.xhtml?persistentId=doi:10.7910/DVN/SJTRBI

Study 3: CC0 license

https://dataverse.harvard.edu/dataset.xhtml?persistentId=doi:10.7910/DVN/WTUGFZ

Study 4: CC BY 3.0 license (Free to remix, transform, and build upon the material for non-profit purpose)

https://data.mendeley.com/datasets/sbmwmsynxk/1

Study 5: CC BY 4.0 license (Free to remix, transform, and build upon the material for any purpose)

https://frontiersin.figshare.com/articles/dataset/Data_Sheet_1_The_Association_Between_Disgust_Sensitivity_and_Negative_Attitudes_Toward_Homosexuality_The_Mediating_Role_of_Moral_Foundations_xls/8234243

Comments from Reviewer 2

Comment: This is one of the empirically strongest papers on the Moral Foundations Questionnaire, with strengths that the paper itself accurately touts. But there are some caveats and weaknesses of which the authors seem unaware.

Response: We thank the reviewer for this fulsome complement.

Comment: The samples are – except for Study 5, the smallest sample – based on predominantly Western respondents (in fact, predominantly Anglo-American). Page 16 claims that study 1 identified a ‘reliable basic structure’ but this might be true only for certain populations. In general, the paper needs more caveats about potential inapplicability to non-Western populations. I don’t think we should be having a few Chinese university attendees representing (standing in for) the entire non-Western world as is done here.

Response: We agree that the lack of global and cultural coverage is a limitation of the study. We now mention this in the Limitations section of the paper’s discussion saying ‘though, given that we used only one non-Western sample, this requires validation” See line 709, page 30

Comment: With respect to the one non-Western sample, the paper fails to note important details (the sample was evidently almost 80% female) and there are some questions. What universities were the respondents from and how Westernized were they? Why was this particular sample (among all non-Western samples administered the MFQ) chosen, and did that involve cherry-picking the sample most likely to be supportive of the select model?

Response: Thank you for pointing this out. In addition to stating that the sample is 80% female in the participants’ section, we now note this in the Limitations section of the paper’s discussion. We say “This sample also had higher proportion of females (78 %) compared to other samples”. See lines 717-718, pages 30-31. We used this sample because it was the only non-western sample with MFQ data available to us.

Comment: One could also wonder if there is some degree of cherry-picked subjects in studies 1-4. The description of the study 1 sample does not rule out that these are all moral-psychology enthusiasts who volunteered. The study 2 was highly motivated – found their way to a website on moral psychology. Study 3 and 4 samples were from MTurk, but might be called ‘cream of the crop’ MTurkers based on the selection criteria; that’s well and good unless it means these results are dependent on using ‘elite questionnaire-response professionals’ and are ungeneralizable to a typical population. What percentage of MTurk participants meet the criteria specified, and were any MTurk participants recruited but then eliminated (in order to make favorable results more likely)? (A general take-home message might be that the intended structure of the MFQ is fragile, hard to locate in your typical noisy data.)

Response: We agree that this is a potential limitation of the study. We have added this as a limitation. We now say in the Limitations section of the paper’s discussion (lines 721-725, page 31:

“It is also worth mentioning that in Study 2 we used data provided by participants who voluntarily visited a moral psychology website and, presumably, were highly motivated. Similarly, in Study 3-4 we used data provided only by MTurk participants with high approval rate. This may have affected how representative of general population these samples are”. 

Comment: The models employed are entered in a particular order that was apparently set a priori (is there a pre-registration to verify this?), but the question is raised whether, when all is said and done, one (or more) of them might be unnecessary. That is, perhaps by backward removal of the element adding least, a more parsimonious outcome might be reached. This is analogous to the forward versus backward methods in stepwise regression. Related questions: How does it make sense to enter a hierarchical model (which includes the five-factor model) before one enters the five-factor model, and what is the difference between ‘two-factor model’ and ‘binding and individualizing’ foundations in, for example, Table 8.

Response: Thank you for this suggestion about parsimony and reducing a five-factor model rather than adding to it. Reducing the five-factor model to smaller number of factors made model’s fit worse, not better, we also added a sentence about our approach, which is “Our approach was to develop models increasing in complexity from the simplest predicted model, to more complex structures, as required to achieve good fit”. See page 11, lines 248-250 We did not, unfortunately pre-register this method. 

Comment: Following that same thread: The conclusion (though unstated) would seem to be that one can use the MFQ, but must really analyze it or make sense of its scores in a complicated and onerous manner.

Response: We include a scoring system at item level and sharing the models which can be used to generate scores, available at OSF page of the paper. However, we agree that it would desirable to add items to balance the new scales.

Comment: That is perhaps a problem for construct validity. An important area of future implications would be this: What do these results suggest about how to construct a better moral ‘foundations’ measure (e.g., one that would not be so profoundly affected by method variance, or by the pull of an underlying two-factor model and of political leanings and of acquiescence or general morality vs. amorality? In other words, how can the measurement of moral foundations be cleaned up in these regards?

Response: Thank you for this suggestion, we now include very similar wording in the modelling implications section (See lines 691-693, pages 29-30 and lines 609-612, page 26): 

“Constructing a better measure of moral foundations may also require rephrasing of some items and perhaps creating new ones to remove or at least minimize the effects of method variance and possible bias” 

and

“A future theory of moral judgment will need to propose and test hypotheses about the selective pressures on loyalty to country vs loyalty to kin and purity vs sanctity factors. In addition, constructing a modified measure of moral foundations may require generating new items to cover the larger number of domains.”.

Smaller matters:

Comment 7: The middle paragraph on page 26 leads to considerable head-scratching. The first sentence is incomprehensible. As for the 2nd, why is social dominance picked out so prominently among all possible alternatives? What is the ‘alignment’ referred to there? It would seem that one interpretation of the results is that ‘tilt’ and binding-individualizing make independent contributions because one cannot reduce the latter entirely to tilt (as some treatments of the topic have implied in the past), there being both liberal and conservative ways of endorsing binding AND of endorsing individualizing morality.

Response: Thank you for this correction. We removed first sentence and provided a reference for the possible link between the tilt and social dominance. We say: 

“Future study could usefully focus on understanding this factor, including relating it to other constructs, for instance one or other component of the social dominance construct [48], which was previously offered as an explanation of liberal-conservative differences [56]”. See lines 642-645, page 28.

Comment: In Table 8 it is noteworthy that moving from model 3 to model 4 yielded a huge improvement in fit, which was not the case in other samples. Does this say something about how Chinese respondents characteristically handle morality or this questionnaire?

Response: Thank you for pointing this out. The improvement from model 3 to model 4 (adding Judgment and Relevance method factors) was present in all samples. One explanation why it was more pronounced in the Chinese sample could be the way self-reflection (relevance items) and evaluative judgments (judgment items) are presented in the Chinese language, however since it was the only non-western sample we decided not to speculate about the cause of this effect. We mentioned this anomaly in the discussion, saying about this sample “It also showed the largest model fit improvement by modelling two method factors” see lines 718-719, page 31.

Comment: Page 13 identifies a set of three MFQ items and labels them as sanctity, but it is hard to apply this interpretation to the third item mentioned (about roles for men and women). One could just as easily label this factor as ‘traditional gender roles/expectations’ (that also would fit 2 of the 3 items). 

Response: We agree with the reviewer that the alternative name for this factor is possible, however, we chose to keep the term sanctity as this is the usage from Haidt, in which both gender roles and religion are sanctified, e.g. “If the body is a temple housing divinity within, then people should not be free to use their bodies in any way they please; rather, moral regulations should help people to control themselves and avoid sin and spiritual pollution in matters related to sexuality, food, and religious law more generally” (Haidt and Graham, 2007).

Comment: Page 13-14 separate out a patriotism factor from a loyalty factor, but it would seem more informative to label one of them as loyalty to country and the other as loyalty to family/team/group (i.e., to smaller-scope entities).

Response: Thank you for this suggestion. We have re-worked the discussion for study 1 and ongoing discussion to pay close attention to clarity in terminology and adopted a clearer naming scheme. We say (see lines 420-424, page 18): 

“This seven (rather than five) foundation model broke-out items from the sanctity/purity, authority/respect, and ingroup/loyalty foundations to form independent sanctity and purity foundations, and independent foundations of loyalty to country and loyalty to clan. These changes also altered the nature of the authority foundation, leaving it more obviously aligned around hierarchy.”

 See lines 420-424, page 18. We have also added a figure showing the mapping from the 5 hyphenated original foundations to the 7-factor model. (Figure 6).

Comment: In reference to same on page 24, it is implied that Herbert Simon differentiated patriotism from loyalty, but this seems unlikely given how Simon’s viewpoint is stated. Clarity needed.

Response: Thank you for pointing this out. We have clarified that this refers to hierarchy. We say (see lines 599-605, page 26) 

“This distinction of loyalty to one’s group and obedience to hierarchy, has a distinguished history in theory. In his work on administrative behavior, Simon [54] identified loyalty and obedience as the two necessary conditions for the existence of organisations, defining loyalty as the capacity to introject organizational objectives in place of one’s own aims and obedience as choosing to make one’s default response be to follow requests of a superior: a definition which corresponds closely to notions of respecting the wishes of those in authority.”

We also call this factor (consisting of 4 Authority items) as “hierarchy” throughout in the paper.

Comment: Until the very last part of the paper, there is a tendency to label one latent variable -- created by constraining all items to load positively on it – as General Morality although an equally plausible interpretation is Acquiescence. Would be better to mention both possible interpretations from the beginning.

Response: Thank you for the suggestion. We now call this general/acquiescence factor throughout in the paper.

Comment: In Table 1, the “(modified)” notation for the best-fitting model for the Lurino and Saucier (2018) paper needs some explanation.

Response: Thank you for pointing this out. We now explain “modified” in the table’s footnote. We say: “Iurino & Saucier (2018) used alternative model derived from exploratory factor analysis of original MFQ items”. See Table 1, page 8, 

Comment: The tendency to overestimate one’s personality traits is (on p. 12) mischaracterized as halo effect. Halo effect is more commonly used to refer to overestimation of someone else’s positive traits. Doing it to yourself is self-enhancement or social desirability bias.

Response: Thank you for this correction. We now describe these as “self-enhancement or social desirability”. See line 302-303, page 13.

Comment: It seems that some text on pages 17-18 is redundant with what was said before in the paper, this needs a check.

Response: Thank you for pointing this out. We removed the redundant text.

---

## [Decision Letter · Decision Letter 1]

8 Oct 2021

Remapping the foundations of morality: Well-fitting structural model of the Moral Foundations Questionnaire.

PONE-D-21-17063R1

Dear Dr. Zakharin,

We’re pleased to inform you that your manuscript has been judged scientifically suitable for publication and will be formally accepted for publication once it meets all outstanding technical requirements.

Kind regards,

Peter Karl Jonason

Academic Editor

PLOS ONE

Additional Editor Comments (optional):

Reviewers' comments:

Reviewer's Responses to Questions

**Comments to the Author**

1. If the authors have adequately addressed your comments raised in a previous round of review and you feel that this manuscript is now acceptable for publication, you may indicate that here to bypass the “Comments to the Author” section, enter your conflict of interest statement in the “Confidential to Editor” section, and submit your "Accept" recommendation.

Reviewer #1: (No Response)

2. Is the manuscript technically sound, and do the data support the conclusions?

Reviewer #1: Yes

3. Has the statistical analysis been performed appropriately and rigorously? 

Reviewer #1: Yes

4. Have the authors made all data underlying the findings in their manuscript fully available?

Reviewer #1: Yes

5. Is the manuscript presented in an intelligible fashion and written in standard English?

Reviewer #1: Yes

6. Review Comments to the Author

Reviewer #1: First, well done on asking to use other’s data; Kevin let me know you did. I’d suggest you list the grants that funded all of the data you used in the acknowledgements; whether you wish to thank the PI’s is of course up to you.

I consider reviews to be suggestive. Ultimately it is up the authors to decide what to put in their papers. I only give hard rejects when the analyses or understanding of the literature is so wrong, that there is no hope for meaningful or valid contribution.

Overall, I’m supportive pf publication, as I was originally. That said, I am disappointed in the minimal revision made and I think the paper undersells some major points. So yes publish, it is a fine empirical paper. Whether you want it to be a better paper, that’s always a choice between investing more time vs. just get it out it.

I do think it is a missed opportunity to not engage Haidt’s theory here. If your main takeaway is that the factor structure of the MFQ is not the 5 dimensions that Haidt argues for, and the items don’t fit as advertised into their subdimensions, then you achieve that. But despite valiant efforts, this means that half of MFT is not supported by the measures or data. This seems a rather important point and one the paper appears unwilling to engage. If the factor structure is not valid, then the theory or measures are not valid. As it stands there is now a mismatch between MFT and MFQ. The current paper sidesteps this question, but in my view this should be the paper to actually engage it. A single sentence, of let someone else do it, seems both a missed opportunity and also a sin- well Lindon would say that’s too strong a word, and I agree but I don’t have his vocabulary, so going with it. But I’ll ask you this, what other paper would this be addressed in? Empiricism without theoretical segment has value but is limited. Read the book, go at it straight on, remains my suggestion.

7. PLOS authors have the option to publish the peer review history of their article (what does this mean?). If published, this will include your full peer review and any attached files.

Reviewer #1: No

---

## [Editor Report · Acceptance letter]

13 Oct 2021

PONE-D-21-17063R1 

Remapping the foundations of morality: Well-fitting structural model of the Moral Foundations Questionnaire. 

Dear Dr. Zakharin:

I'm pleased to inform you that your manuscript has been deemed suitable for publication in PLOS ONE. Congratulations! Your manuscript is now with our production department. 

Kind regards, 

on behalf of

Dr. Peter Karl Jonason 

Academic Editor

PLOS ONE